# Evaluating Neuron Interpretation Methods of NLP Models

**Yimin Fan**[♡]    **Fahim Dalvi**[◇]    **Nadir Durrani**[◇]    **Hassan Sajjad**[♣]
[♡]The Chinese University of Hong Kong, Hong Kong, China
[◇]Qatar Computing Research Institute, HBKU, Qatar
[♣]Faculty of Computer Science, Dalhousie University, Canada

## Abstract

Neuron interpretation offers valuable insights into how knowledge is structured within a deep neural network model. While a number of neuron interpretation methods have been proposed in the literature, the field lacks a comprehensive comparison among these methods. This gap hampers progress due to the absence of standardized metrics and benchmarks. The commonly used evaluation metric has limitations, and creating ground truth annotations for neurons is impractical. Addressing these challenges, we propose an evaluation framework[1] based on voting theory. Our hypothesis posits that neurons consistently identified by different methods carry more significant information. We rigorously assess our framework across a diverse array of neuron interpretation methods. Notable findings include: i) despite the theoretical differences among the methods, neuron ranking methods share over 60% of their rankings when identifying salient neurons, ii) the neuron interpretation methods are most sensitive to the last layer representations, iii) Probeless neuron ranking emerges as the most consistent method.

## 1   Introduction

The advent of deep neural network (DNN) models and their success in Natural Language Processing (NLP) has opened a new research frontier to interpret the representations learned within these models. Representation analysis provides a holistic view of the knowledge learned within the representation. Whereas neuron analysis gives insight into how knowledge is structured within the representation. To this end, an ample amount of work has been done to understand the knowledge captured within the learned representations (Belinkov et al., 2017; Liu et al., 2019a; Tenney et al., 2019; Dalvi et al., 2022) and individual neurons (Karpathy et al., 2015; Kádár et al., 2017a; Dalvi et al., 2019).

Neuron analysis provides a fine-grained interpretation of the representation. It discovers salient neurons of the network w.r.t to a concept such as morphological or syntactic concepts (Lakretz et al., 2019; Durrani et al., 2020), and provides insight into how they may be used during inference (Lundberg & Lee, 2017; Dhamdhere et al., 2018). Beyond understanding the inner dynamics of the DNN models, neuron analysis has various potential applications such as network manipulation (Bau et al., 2019), domain adaptation (Antverg et al., 2022), model distillation (Dalvi et al., 2020) and architectural search (Prasanna et al., 2020).

A number of neuron analysis methods have been proposed in the literature to interpret deep NLP models. While these methods have shown promising results, there is a notable absence of systematic comparisons among them in existing studies. Conducting a fair comparison of neuron interpretation methods is crucial for comprehending the merits and limitations of each technique. Such comparisons

---

[1]https://github.com/fdalvi/neuron-comparative-analysis

37th Conference on Neural Information Processing Systems (NeurIPS 2023).

serve as a guiding light for the community, steering efforts towards the development of improved and more effective interpretation methodologies.

In this work, we study a variety of neuron interpretation methods under a controlled setting and provide a thorough comparative analysis. There are two main challenges in comparing neuron interpretation methods: i) absence of a *standard evaluation metric*, ii) lack of an *evaluation benchmark* (Sajjad et al., 2022b). The most common evaluation metric used in the neuron analysis literature is to train a classifier using the discovered neurons as features. The performance of the classifier in predicting the concept of interest serves as evidence of the quality of the discovered neurons, in other words, the correctness of the neuron interpretation method.

We argue that the classifier as an evaluation metric is suboptimal for two reasons: i) training a classifier provides an opportunity to learn the task and it may result in good performance irrespective of the quality of the neurons used as features (Hewitt & Liang, 2019), ii) the classifier as an evaluation method may favor the neuron interpretation methods that are methodologically similar (for instance, methods that use a classifier to discover salient neurons). Antverg & Belinkov (2021) shed light on the latter issue and showed that the methodological similarity between the neuron interpretation methods and the evaluation methods may result in unreasonably high accuracy scores. We empirically demonstrate the issue in Section 3. Another bottleneck in comparing neuron interpretation methods is the absence of *evaluation benchmark*. Neurons are distributive in nature and exhibit polysemy (capturing multiple concepts at the same time). Moreover, there can be different sets of neurons learning a concept, and there exists no single correct answer. It is therefore intricately challenging to create the ground truth annotations and infeasible to scale to a large set of models and concepts.

Given the lack of a standard evaluation metric and the infeasibility of creating ground truth annotations, *how can we compare neuron interpretation methods?* We rely on the voting theory (O'Connor & Robertson, 2003) that performs a systematic aggregation of results in order to achieve a consensus (Patrão Neves, 2016). It is effectively used in political, social, and machine-learning communities for cases wherein there are distinct preferences among voters (models). Examples of specific applications of voting theory include ranking search results obtained from various algorithms in information retrieval (Lin et al., 2017; Yilmaz et al., 2008), finding the best machine translation system using pair-wise ranking (Lapata, 2006) and aggregating results of several NLP tasks (Colombo et al., 2022).

We hypothesize that *neurons that are commonly discovered by different interpretation methods are more informative than others*, and may serve as a signal of correctness for the selected neurons. Motivated by this, we devise two voting-based compatibility metrics. Our first metric, `AvgOverlap`, is based on plurality voting (Cooper & Zillante, 2012). `AvgOverlap` computes an average overlap of the discovered neurons across all voters. Our second method, `NeuronVote`, inspired by Borda Count (Lippman, 2012), takes into account the ranking of neurons of each method while calculating the average pair-wise intersection over union with all the other methods. The compatibility metrics provide a single score for each method, where a higher score refers to a neuron interpretation method whose discovered neurons are more compatible with the other methods. We further extend the evaluation of neuron interpretation methods to pair-wise comparison, providing insights into how any two given methods relate in terms of their discovered neurons.

We conduct a comparative analysis of six neuron interpretation methods using diverse concepts consisting of morphological, semantic, and syntactic properties and using three pre-trained models. The analysis suggests that the best performing methods, despite their methodological differences, share more than 60% of the top neurons. Probeless ranking Antverg & Belinkov (2022) consistently exhibited the highest overlap with other methods. Because neurons in the last layer representations are highly correlated, they are the most challenging to interpret. Finally, we present a case study that demonstrates the usefulness of our evaluation methodology for any new method proposed in the future (Appendix 7.1). Our contributions are as follows:

- We conduct the first thorough comparative analysis of a large set of neuron interpretation methods of NLP across 3 pre-trained models and using a diverse set of linguistic concepts.
- Our proposed evaluation methodology consisting of two compatibility metrics and a pair-wise comparison targets one of the critical limitations of the neuron interpretation studies.
- We provide the evaluation methodology as a framework to facilitate future studies and comparison of methods.

## 2 Neuron Interpretation Methods

The goal of neuron interpretation methods is to rank $\mathcal{N}$ neurons with respect to some concept $\mathcal{C}$. These methods can be broadly classified into two classes: corpus-based methods and probing-based methods (Sajjad et al., 2022b). The former discovers the role of neurons by aggregating statistics over neuron activations and the latter trains classifiers to achieve the same.

Let $\mathcal{D}$ be a dataset of sentences (represented as a list of tokens), where each token $w$ appears in a specific context/sentence and has a contextual representation associated with it. Let $z(n, w)$ be the activation of the $n^{th}$ neuron and the $w^{th}$ token in $\mathcal{D}$. The neuron $n \in \mathcal{N}$ can be from any component of the original model, such as a specific layer. Also note that the activation value $z(n, w)$ in context-dependent, so the same $w$ can result in different $z(n, w)$ values depending on context.

A concept $\mathcal{C}$ represents a property we want to discover neurons for and $\hat{\mathcal{C}}$ represents a *random* concept, i.e. containing words that are not associated with the concept $\mathcal{C}$. For example, $\mathcal{C}_{country} =$ {Rome, Paris, London, New York, ... } represent a concept of country names and $\hat{\mathcal{C}}$ would consist of all other words in the corpus that are not country names. More formally, $\hat{\mathcal{C}} = \mathcal{D} \setminus \mathcal{C}$.

Let $R(n, \mathcal{C})$ represents the score of a neuron $n$ with respect to a concept $C$. Each neuron interpretation method implements $R$, and provides a ranked list of neurons with respect to a concept. In the following, we present six interpretation methods[2] studied in this paper.

### 2.1 Corpus-based Methods

Corpus-based methods align neurons with a concept by accumulating co-occurrence patterns between neuron activations and the presence of the concept of interest. We use two corpus-based methods in this work presented as follows.

#### 2.1.1 Probeless

The Probeless method (Antverg & Belinkov, 2021) obtains neuron rankings based on an accumulative strategy. The score of a given neuron $n$ is defined as follows:

$$R(n, \mathcal{C}) = \mu(\mathcal{C}) - \mu(\hat{\mathcal{C}}) \tag{1}$$

where $\mu(\mathcal{C})$ is the average of activations $z(n, w), w \in \mathcal{C}$. $\mu(\hat{\mathcal{C}})$ is the average of activations over the random concept $\hat{\mathcal{C}}$. Note that the ranking for each neuron $n$ is computed independently.

#### 2.1.2 IoU Method

Mu & Andreas (2020) proposed IoU to generate compositional explanations for neurons. For each token in the dataset, they create i) a binary mask of a neuron by thresholding activations above a percentile and ii) a binary mask of a concept by checking its presence in the token. They compute Intersection over Union (IoU) over the two masks:

$$R(n, \mathcal{C}) = \frac{\sum_w \mathbb{1}(z(n, w) > \delta) \wedge \mathbb{1}(w \in \mathcal{C})}{\sum_w \mathbb{1}(z(n, w) > \delta) \vee \mathbb{1}(w \in \mathcal{C})} \tag{2}$$

where $\mathbb{1}$ is the indicator function. $z(n, w) > \delta$ is a binary mask over neuron activations based on the threshold $\delta$. $w \in \mathcal{C}$ is a binary mask, created by checking if the current word represents the concept $\mathcal{C}$ of interest. Mu & Andreas (2020) used IoU scores to generate alternative explanations for neurons on the task of image classification. Here we apply their method in the NLP domain to generate neuron ranking for any concept of interest. Like Probeless, the ranking for every neuron is computed independently of other neurons.

### 2.2 Probing Methods

Probing methods train a classifier using neuron activations as features for the task of predicting the concept of interest (Belinkov et al., 2017; Hupkes et al., 2018). The internals of the trained classifier (e.g. its weights) are used to rank the features (neurons).

---

[2]Available in the NeuroX toolkit Dalvi et al. (2023).

### 2.2.1 Lasso Regularizer (L1)

Radford et al. (2019) trained a linear classifier with L1 regularization and used the weights of the classifier as a proxy for the importance of neurons for the given concept. The loss function of the classifier is as follows:

$$\mathcal{L}(\theta) = -\sum_w \log P_\theta(c|z_\mathcal{N}(w)) + \lambda_1 \|\theta\|_1 \tag{3}$$

where, $c \in \{\mathcal{C}, \hat{\mathcal{C}}\}$ is the correct class for $w$, $z_\mathcal{N}(w)$ is the vector of all neuron activations for word $w$, i.e. $[z(0,w), z(1,w), ..., z(n,w)]$, and $P_\theta(c|z_\mathcal{N}(w))$ is the probability that word $w$ belongs to the class $c$. The learned weights $\theta$ serve as the ranking of neurons. Specifically, $R(n, \mathcal{C})$ is defined as the absolute weights for the class i.e. $|\theta_\mathcal{C}(n)|$.

L1 adds the "absolute value of magnitude" of the coefficient as a penalty term to the $L(\theta)$, which shrinks the less important weights to zero, leading to sparse weight distribution. Radford et al. (2019) used this regularization to force the classifier to learn spiky weights, indicating the selection of a few specialized neurons learning a concept, while setting the majority of neurons' weight to zero. Such an assumption is useful in discovering focused neurons that learned one particular concept only.

### 2.2.2 Ridge Regularizer (L2)

Lakretz et al. (2019) used L2 regularization to train the linear classifier. The loss is as follows:

$$\mathcal{L}(\theta) = -\sum_w \log P_\theta(c|z_\mathcal{N}(w)) + \lambda_2 \|\theta\|_2^2 \tag{4}$$

L2 forces weights to be close to zero (but not zero). It is useful to deal with multicollinearity (neurons that are highly correlated) scenarios, through constricting the coefficient while keeping all the features. Intuitively, this encourages grouping of features, thus discovering group neurons that jointly learn a concept. The score $R(n, \mathcal{C})$ is computed in a similar fashion as the Lasso Regularizer method.

### 2.2.3 ElasticNet Regularizier (LCA)

Both L1 and L2 regularizations capture properties that are desirable when selecting the most important neurons. L1 facilitates sparsity, identifying focused neurons while L2 encourages identifying groups of highly correlated features into account. Dalvi et al. (2019) used ElasticNet regularization (Zou & Hastie, 2005) that balances the trade-off between them.

$$\mathcal{L}(\theta) = -\sum_w \log P_\theta(c|z_\mathcal{N}(w)) + \lambda_1 \|\theta\|_1 + \lambda_2 \|\theta\|_2^2 \tag{5}$$

$\lambda_1$ and $\lambda_2$ are hyperparameters that are tuned to optimize the effect of L1 and L2 regularization.

### 2.2.4 Gaussian Classifier

Hennigen et al. (2020) assumes that neurons exhibit a Gaussian distribution. They fit a multivariate Gaussian, say $\mathcal{P}$ over all neurons. Since multivariate Gaussian is decomposable by nature, they are able to extract individual probes for any subset of input features without additional work. Formally, they are able to extract $\mathcal{P}_\mathcal{F}$, where $\mathcal{F} \subset \mathcal{N}$. The classifier itself is trained using the Bayesian framework, specifically the maximum a posteriori estimate to compute the parameters $\theta$. Once a multivariate Gaussian is trained, the neuron selection is performed in a greedy fashion:

$$F = (), F = F \oplus \arg\max_n \mathcal{P}(\mathcal{C}|F \oplus \{n\}) \quad \text{for } i \text{ in } 1..|\mathcal{N}|$$
$$R(n, \mathcal{C}) = |\mathcal{N}| - index(F, n) \tag{6}$$

In essence, probes for individual neurons are first extracted, and the neuron probe with the highest log-likelihood is considered the most important neuron. Next, probes are extracted for pairs of neurons, with one of them being the selected neuron. The pair with the highest log-likelihood then contributes the second most important neuron to the ranking. The full ranking is compared iteratively, with each step being a greedy selection of the next best neuron to add based on the log-likelihood. This method makes two fundamental assumptions: i) neuron activations follow a Gaussian distribution and ii) neuron ranking can be done in a greedy fashion.

# 3 Evaluation Methodology

## 3.1 Classifier Accuracy

The commonly used evaluation metric i.e. classification accuracy of the selected neurons is inadequate: i) because it favors the neuron ranking methods that are based on the probing framework (Antverg & Belinkov, 2021), ii) it is not clear whether the classifier performance is a reflection of the knowledge learned in the discovered neurons or it is due to the capacity of the classifier to learn and memorize the task (Hewitt & Liang, 2019; Zhang & Bowman, 2018).

We demonstrate these issues empirically, by using a linear classifier without regularization as an evaluation metric to evaluate various neuron interpretation methods. Given $s$ discovered neurons with respect to a concept $\mathcal{C}$, we train the classifier using these discovered neurons as features. The performance of the classifier serves as a measure of the correctness of $s$. Table 1 presents the average accuracy across all parts of speech (POS) concepts and across all layers of BERT for $s = 30, 50, 70, 100$. Random refers to a random selection of neurons. The details of the dataset and the experimental setup are provided in Section 4. A few notable observations are: the classifier accuracy is higher for the probing methods (Lasso and Ridge) than the corpus-based methods (Probeless and IoU), ii) the classifier has the capacity to memorize as a Random selection of neurons performs reasonably well when more than a certain number of neurons are used in the evaluation. These results complement the issues raised earlier i.e. the classifier may result in high performance irrespective of the correctness of the neurons, and it favors the interpretation methods that are methodologically similar to itself.

Table 1: *Task: POS, Model: BERT,* Accuracy scores using classifier accuracy as an evaluation metric. Bold and underline are the first and second best results.

| Neurons | 30 | 50 | 70 | 100 |
|---------|------|------|------|------|
| Probeless | 0.92 | 0.94 | 0.95 | 0.96 |
| IoU | 0.91 | 0.93 | 0.94 | 0.95 |
| Lasso | 0.93 | 0.95 | 0.95 | 0.96 |
| Ridge | **0.95** | **0.97** | **0.98** | **0.98** |
| Random | 0.81 | 0.86 | 0.89 | 0.92 |

Table 2: *Task: POS, Model: BERT,* Average compatibility scores `AvgOverlap` and `NeuronVote` when selecting the top 10, 30, 50 neurons from layers 1, 6, 12. Bold and underline are the first and second best scores.

| | AvgOverlap | NeuronVote |
|---------|-----------|-----------|
| Random | 0.021 | 0.021 |
| Gaussian | 0.086 | 0.169 |
| LCA | 0.258 | 0.514 |
| Lasso | 0.240 | 0.473 |
| Ridge | 0.177 | 0.362 |
| Probeless | **0.269** | **0.532** |
| IoU | 0.156 | 0.365 |

## 3.2 Compatibility Metrics

Given the absence of a good evaluation metric and gold annotations, we rely on consensus-based voting theory. The consensus serves as a tool to measure the relation between rankings. In NLP, it has been widely used in information retrieval to combine search results of various algorithms (Lin et al., 2017; Yilmaz et al., 2008), in machine translation to compare different systems' output using pair-wise ranking (Lapata, 2006) and to aggregate results of NLP tasks (Colombo et al., 2022).

One way to compare rankings is to perform pairwise comparison using rank correlation or similarity functions such as Kendall's $\tau$ (Kendall, 1938) and Spearman correlation (Daniel, 1990). These methods assume conjoint rankings where all rankings contain identical elements. In the case of neuron rankings, for every neuron interpretation method, we only consider the $s$ top-ranked neurons with respect to a concept. The rationale behind this setting is to minimize the noise and randomness in rankings that may occur for neurons that are not learning the concept. This results in disjoint sets and the rank correlation methods can not be directly applied in such cases (Lin et al., 2017).

Motivated by the voting theory, we devise two metrics to compare disjoint rankings. Our first metric, `AvgOverlap`, is based on plurality voting (Cooper & Zillante, 2012). `AvgOverlap` computes the pair-wise average neuron overlap across methods. It is a set-based method and it does not consider the internal ranking of neurons into consideration. For example, the first top neuron and fifth top neuron in a ranking will get equal weight when compared with another ranking. Our second method, `NeuronVote`, inspired by Borda Count (Lippman, 2012), considers the order of neurons into consideration in comparing rankings. We call our methods compatibility metrics, which when given

a neuron ranking, provide its compatibility score with respect to other methods. The metrics give a high score to a method that is most aligned with the rankings of the other neuron interpretation methods. To build a deeper understanding of how any two methods relate in terms of the resulting neurons, we also present a pair-wise comparison of the discovered neurons across methods.

Formally, let $S_m = n_1, n_2, ..., n_s$ represent the $s$ top-ranked neurons for a method $m$, where $s$ is a hyperparameter that can be adjusted to vary the number of top neurons selected for evaluation. We describe the compatibility metrics and the pair-wise comparison method as follows.

### 3.2.1  Average Overlap

AvgOverlap scores an interpretation method based on its average neuron overlap with other methods. We define the overlap $o(m_1, m_2)$ between two methods as the intersection over union of their respective top-neuron sets:

$$o(m_1, m_2) = \frac{|S_{m_1} \wedge S_{m_2}|}{|S_{m_1} \vee S_{m_2}|} \tag{7}$$

AvgOverlap is defined as:

$$\text{AvgOverlap}_{m_i} = \frac{1}{\mathcal{M} - 1} \sum_{j=1, j \neq i}^{\mathcal{M}} o(m_i, m_j) \tag{8}$$

where $\mathcal{M}$ is the set of all methods. A large value of AvgOverlap means that the ranking of that method is well aligned with the ranking of other neuron interpretation methods.

### 3.2.2  NeuronVote

AvgOverlap treats each neuron equally in a set of discovered neurons and ignores the ranking assigned by the methods. However, intuitively a neuron interpretation method whose top choice is endorsed by all other methods should get a higher score than an interpretation method whose 10th top choice is endorsed by all other methods. To take this into account, we follow the Borda count strategy (Lippman, 2012) and propose NeuronVote. It considers each neuron's ranking in the voting process. We first create $S_{best}$, an aggregated ranking based on the position of individual neurons in the rankings of various methods.

$$S_{best} = argsort \left( \sum_m^{\mathcal{M}} s - index(S_m, n) \quad \text{for } n \in \mathcal{N} \right) \tag{9}$$

where $argsort$ is an ascending sort function and $index$ gives the position of neuron $n$ in the ranked list. The accumulated $S_{best}$ represents an aggregation on weighted votes, with neurons appearing at the top of various methods' ranked lists getting a higher weight than others. Following the notation in AvgOverlap, we define $\text{NeuronVote}_{m_i} = o(S_{m_i}, S_{best})$. The method with high NeuronVote implies that its ranking of neurons is most endorsed by other interpretation methods.

### 3.3  Pairwise Comparison

The compatibility metrics provide a holistic evaluation of a neuron interpretation method with respect to other methods. We further extend the analysis to pair-wise comparison to understand how the discovered neurons of a method related to another method. The pair-wise comparison is calculated as an intersection between the output of two methods i.e. $o(m_1, m_2) = S_{m_1} \wedge S_{m_2}$

## 4  Evaluation

### 4.1  Settings

We consider three 12-layered pre-trained models: BERT-base-cased (BERT, Devlin et al., 2019), RoBERTa-base-cased (RoBERTa, Liu et al., 2019b) and XLM-Roberta-base (XLMR, Conneau et al., 2019). Each layer consists of 768 neurons. We consider concepts from three linguistic tasks: parts of speech tags (POS, Marcus et al., 1993), semantic tags (SEM, Abzianidze et al., 2017) and syntactic chunking (Chunking) using CoNLL 2000 shared task dataset (Tjong Kim Sang & Buchholz, 2000). Dataset details are provided in Appendix 7.2. We consider every tag as a concept and identify neurons with respect to the concept. In the case of classifier-based methods, we use a binary classification setup where the contextualized representation of words belonging to the concept serves as positive

Table 3: *Task: POS,* Average `NeuronVote` compatibility scores across concepts when selecting the top 10, 30, and 50 neurons from layers 1, 6 and 12. Bold numbers, underline numbers, and dashed numbers show the first, second, and third best scores respectively

| | BERT | | | RoBERTa | | | XLMR | | |
| Layers | 1 | 6 | 12 | 1 | 6 | 12 | 1 | 6 | 12 |
|---|---|---|---|---|---|---|---|---|---|
| Random | 0.019 | 0.021 | 0.023 | 0.020 | 0.019 | 0.017 | 0.020 | 0.017 | 0.023 |
| Gaussian | 0.147 | 0.177 | 0.183 | 0.140 | 0.179 | 0.179 | 0.118 | 0.135 | 0.153 |
| LCA | **0.501** | 0.544 | 0.391 | 0.550 | 0.495 | 0.380 | 0.450 | 0.531 | 0.373 |
| Lasso | 0.496 | 0.545 | 0.395 | 0.475 | 0.453 | 0.350 | 0.410 | 0.471 | 0.267 |
| Ridge | 0.332 | 0.405 | 0.360 | 0.452 | 0.491 | 0.510 | 0.358 | 0.446 | 0.455 |
| Probeless | 0.497 | **0.550** | **0.515** | **0.590** | 0.589 | **0.580** | **0.534** | **0.590** | **0.497** |
| IoU | 0.343 | 0.380 | 0.348 | 0.315 | 0.338 | 0.275 | 0.334 | 0.320 | 0.344 |

examples. We randomly select the negative class words from the rest of the data, equal in size to the positive class examples. We drop the concepts that have less than 200 examples to ensure stable training. We split the binary classification dataset into train/dev/test splits of 70/15/15 percent. We use $\lambda_1 = 0.01$ and $\lambda_2 = 0.01$ for regularization-based methods. We present the results of POS only in the main paper and provide the rest in Appx. 7.3.

**Number of top neurons:** We did not optimize the number of neurons $s$ selected by each method as this would result in a varying number of neurons and will make the compatibility score incompatible across methods. We consider $s = 10, 30, 50$ which covers a diverse range to generalize the findings.

**Baseline:** We generate a random ranking of neurons to compare the behavior of a neuron interpretation method with a random selection of neurons. We refer to it as *Random*.

**Procedure:** We extract layer-wise contextualized representations of words in the dataset. For every layer representation, we discover the top neurons with respect to a concept using each neuron interpretation method. We follow a leave-one-out strategy to calculate the compatibility score of each method with respect to other methods. More specifically, we consider one neuron interpretation method as a test method and select the rest as the database of the sets of top neurons. We then calculate the `AvgOverlap` and `NeuronVote` scores of the test method.

### 4.2   Compatibility Scores

Table 2 presents the average compatibility scores across all concepts and when selecting the top 10, 30, and 50 neurons from layers 1, 6, and 12.

**`AvgOverlap` and `NeuronVote` show a similar pattern:** While the scale of the compatibility score is different between both metrics, they are consistent in highlighting the top interpretation methods i.e. Lasso, LCA, and Probeless. In the rest of the paper, we mainly present the results of `NeuronVote`, and discuss the `AvgOverlap` only when it is necessary to show a different trend.

**Overall comparison across methods:** Table 3 presents the `NeuronVote` scores averaged over all POS concepts and when selecting the top 10, 30, and 50 neurons. The bold, underline, and dashed numbers represent the first, second, and third best compatibility scores respectively. **Probeless consistently achieves the highest compatibility across all layers and models.** In other words, the neurons selected by Probeless have the highest overlap with the neurons selected by other methods. We did not observe a similar trend for IoU which is also a corpus-based method.

**LCA and Lasso achieve the second-best compatibility score** (see underlined and dashed numbers in Table 3). Ridge, although methodologically similar to LCA and Lasso, did not consistently show high compatibility across models. We discuss these methods later in Section 4.3. Gaussian achieved the worst compatibility score which is lower by 0.354 points from the best `NeuronVote` score on layer 1 and is only 0.128 better than Random for BERT. The result of Gaussian presents an interesting scenario where the compatibility score suggests that the neurons identified by Gaussian are least similar to all other methods. However, the difference with Random, although small, suggests that the selection of neurons is not random. The low results of Gaussian are inline with Antverg & Belinkov (2021) where they found that the Gaussian method memorizes the probing task and it may not provide the most faithful ranking of neurons. These scores, therefore, suggest that further investigation may be required to confirm the efficacy of the Gaussian method.

A notable point about the top 3 methods (Probeless, LCA, and Lasso) is their methodological diversity, despite of which they selected similar neurons to other methods. On the other hand, methodologically similar techniques such as IoU and Probeless did not result in similar compatibility scores. This shows that the compatibility metric is not biased by methodological similarity, which was one of the primary concerns with the commonly used evaluation metrics.

**Layer-wise trend:** Ethayarajh (2019) showed that each layer is different in terms of representation geometry. Sajjad et al. (2022a) revealed that due to the geometry of the representation space, particularly in the last layers, knowledge of a concept may not be readily available. We hypothesize that the performance of certain neuron interpretation methods may vary based on the nature of the representation space. Our compatibility methods facilitate quantitative evidence to support this hypothesis by showing that certain neuron analysis methods suffer in selecting the most compatible neurons from the higher layers. LCA and Lasso both showed a substantial drop in their compatibility scores for the 12th layer (for example, Table 3: `NeuronVote` LCA – 0.544 for layer 6 and 0.391 for layer 12 for BERT). In contrast, Probeless did not show any substantial difference in the scores of the last layers and earlier layers (`NeuronVote`: Probeless – 0.550 vs 0.515 for layer 6 and 12 respectively for BERT). This further support Probeless as the most reliable method and encourages further investigation into the robustness of Lasso and LCA with varying representational space.

**Compatibility across models:** The overall trend of top-performing neuron interpretation methods is similar across BERT, RoBERTa, and XLMR. One notable difference is the compatibility score of Ridge for the last layer of the RoBERTa (0.510) and XLM-R (0.439) models which is substantially higher compared to the BERT model (0.360). Moreover, it achieved the second best score for XLM-R after Probeless and is substantially better than LCA and Lasso. Ridge prefers correlated features in contrast to Lasso, which prefers a few spiky features highly predictive of the concept. Based on the results, we hypothesize that the last layers of RoBERTa and XLM-R consist of highly correlated neurons and Ridge is effective in discovering salient neurons of correlated nature. LCA balances between spiky and correlated neurons, and results in higher scores than Lasso in these cases.

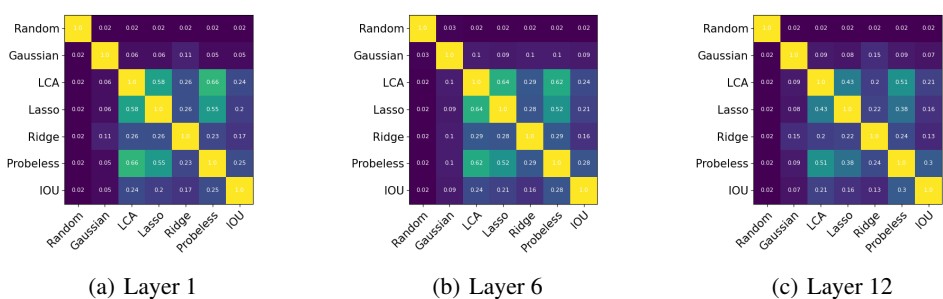

| (a) Layer 1 | (b) Layer 6 | (c) Layer 12 |

Figure 1: *Task: POS, Model: BERT,* Comparing average overlap of top 10-50 neurons

**Random selection of neurons:** The compatibility score of Random serves as a baseline that provides a perspective of the correctness of a newly proposed neuron interpretation method. It gave substantially lower scores than all neuron interpretation methods (Table 3). Even though the Gaussian showed the lowest compatibility score among interpretation methods, the difference with Random adds a factor of confidence to its ranking. Moreover, a low compatibility score for Random also implies that each of the other methods are bringing in important neurons to some degree.

## 4.3 Pairwise Comparison

The compatibility metrics provide a single score to understand an interpretation method in the context of a large set of other interpretation methods. We further extend the analysis to pair-wise comparison and provide insights into the relationship between different methods. Figure 1 presents the heatmaps comparing the average overlap of the top 10 – 50 neurons across methods using BERT. The light color in the heatmap shows higher matches while the dark color shows lower matches. We added *Random* – a random selection of neurons as a baseline. It showed an average overlap of less than 3% only (see the first row in heatmaps).

**Comparing Regularization-based Methods:** LCA and Lasso showed a high overlap of up to 64% in the discovered neurons (see Figure 1(a), 1(b)). However, Ridge results in a lower overlap of at most

29% compared to other methods. Note that the Ridge regularization does not force the coefficients of neurons to zero, thus considers most of the neurons during the training phase. The resulting top neurons may represent correlated neurons capturing similar and redundant knowledge (Dalvi et al., 2020) and may not be a representative of the neurons with the most predictive power with respect to the understudied concept. Lasso prefers individual neurons most predictive of the concept and ElasticNet (LCA) provides a good balance between selecting individual neurons and correlated neurons highly predictive of the concept.

**Corpus-based vs. Regularization-based Methods:**  Probeless and IoU both directly rely on the activation value of a neuron. However, we did not observe a consistent overlap between their discovered neurons (at most 30% in Figure1(c)). In comparison to other classes of methods, Probeless showed a better and consistent overlap with LCA and Lasso. For example, in Figure 1(a) we observed an overlap of 66% between Probeless and LCA, and 55% between Probeless and Lasso. The relatively high overlap of Probeless with methods using Lasso points towards the selection of identical focused neurons by these methods despite their methodological differences.

**Outliers:**  Both Ridge and IoU show a smaller but consistent overlap with most of the methods. As discussed earlier, Ridge works well for correlated features which may not be highly predictive of the concept. IoU relies on activation values and it may select neurons that consistently activate higher than the threshold irrespective of the concept. Gaussian did not show substantial overlap with any methods or with any combination of methods. Further investigation is needed to evaluate its' efficacy.

## 5   Related Work

The area of interpreting deep learning models constitutes a broad expanse of research. This section provides a synthesized overview of diverse interpretability subareas within deep learning models applied to Natural Language Processing (NLP), while also outlining the scope of our study.

*Feature importance and attribution methods* identifies the contribution of input features to predictions. These methodologies predominantly rely on the gradient of the output concerning the input feature and determine input feature importance by evaluating the magnitude of gradient values (Denil et al., 2014; Sundararajan et al., 2017; Danilevsky et al., 2020).

*Counterfactual Intervention* revolves around an intricate analysis of the interplay between input features and predictions. This approach involves manipulating inputs and quantifying resulting output alterations. Diverse intervention strategies, including erasing input words and substituting words with different meanings have been used  (Li et al., 2016b; Ribeiro et al., 2018).

*Attention Weights:* Numerous investigations have been directed towards interpreting components of deep learning models at varying levels of granularity. For instance, attention weights have emerged as a viable metric to gauge the interrelation between input instances and model outputs (Martins & Astudillo, 2016; Vig, 2019). Along these lines, Geva et al. (2021) delved into the analysis of feedforward neural network components within the transformer model, revealing their functionality as key-value memories. Additionally, Voita et al. (2019) demonstrated that pruning many attention heads has minimal impact on performance.

*Mechanistic Interpretability* puts a focus on reverse engineering of network weights to comprehend their behavior (Nanda et al., 2023). Building upon the Distill Circuits thread, Elhage et al. (2021) investigated two-layered transformer models with attention blocks, identifying attention heads contributing to in-context learning. This understanding was further extended to larger transformer-based language models by Olsson et al. (2022). To enhance neuron interpretability, Elhage et al. (2022) introduced a Softmax Linear unit as an activation function replacement. Wang et al. (2022a) attempted to bridge mechanistic interpretability findings in small networks to large ones, particularly GPT-2 small. Their approach involved iteratively tracing influential model components from predictions using causal intervention. They showcased the potential of mechanistic interpretability in understanding extensive models, while also highlighting associated challenges. Similarly, Bricken et al. (2023) uses a sparse autoencoder to disentangle polysemantic neurons.

*Representation Analysis* involves probing network representations concerning predefined concepts, to quantify the extent of knowledge captured in these representations (Conneau et al., 2018; Liu et al., 2019a; Tenney et al., 2019; Durrani et al., 2019; Arps et al., 2022). This is often realized through

training diagnostic classifiers for linguistic concepts, wherein classifier accuracy serves as an indicator of concept knowledge within representations. See Belinkov & Glass (2019) for a comprehensive survey.

*Neuron Interpretation* A more intricate form of representation analysis, termed neuron interpretation, delves into how knowledge is structured within the network (Sajjad et al., 2022b). This approach establishes connections between neurons and predefined concepts, offering insights into where and how specific concept knowledge is assimilated. Work done on neuron analysis can be broadly classified into three groups: Neuron visualization involves manual identification of patterns across a set of sentences (Li et al., 2016a; Karpathy et al., 2015). More recently Foote et al. (2023) proposed an automated approach to enhance interpretability of Large Language Models (LLMs) by extracting and visualizing individual neuron behaviors as interpretable graphs. Corpus-based Methods explore the role of a neuron through techniques such as ranking sentences in a corpus (Kádár et al., 2017b), generating synthetic sentences (Poerner et al., 2018) that maximize its activation, or computing neuron-level statistics over a corpus (Mu & Andreas, 2020; Suau et al., 2020; Antverg & Belinkov, 2022). Bills et al. (2023); Mousi et al. (2023) proposed the use of LLM to interpret neurons. Probing Methods identify salient neurons for a concept by training a classifier using neuron activations as features (Radford et al., 2019; Lakretz et al., 2019; Durrani et al., 2022) or fitting a multivariate Gaussian over all neurons and then extracting individual probes for single neurons (Hennigen et al., 2020).

A number of works identify neurons with respect to the output class (Wang et al., 2022b; Dai et al., 2022). They are effective in finding neurons that play a role in the prediction. In this paper, we focus on the neuron interpretation methods that take a concept as input and find neurons with respect to the concept. We propose an evaluation framework to formalize the evaluation and comparison of results across methods. Moreover, we propose a novel method, MeanSelect and present a case study of using the evaluation framework.

# 6    Conclusion and Limitation

We provided a thorough comparative analysis of neuron interpretation methods in NLP. We overcame the challenge of the lack of a standard evaluation metric and gold annotations by proposing an evaluation strategy consisting of two compatibility metrics and a pair-wise comparison. In addition to developing a capability to evaluate a new neuron analysis method, we presented various insights into existing neuron interpretation methods. For example, due to the correlated nature of last layer representations, they are most challenging for interpretation methods. The selection of top neurons overlaps substantially irrespective of the methodological differences among techniques. We made the evaluation framework available to the research community.

**Limitations:** Our methodology relied on consensus and thus inherits the limitations of that paradigm. The consensus may mislead the evaluation under certain settings e.g. if the majority of the voters form a lobby then they can skew the results of the consensus. This scenario can happen if a set of methods always produce close to identical ranking, they will cause a decrease in the compatibility score of a new method that produces a ranking different from theirs. Another possible issue is if a new method discovers an entirely different set of neurons than the ones discovered by other methods. The compatibility score of such a method will be low. We intend to mitigate these issues by including a large number of diverse neuron interpretation methods that are theoretically different from each other. We believe that under these settings, the scenario of an entirely different set of discovered neurons is more theoretical than practical. We further present a pair-wise analysis that provides an analysis at a more granular level and bring insight into how a method compares with other methods. The pairwise comparison heatmap will highlight if a set of methods form a lobby.

Another limitation of current neuron interpretation methods is that they do not explicitly target the discovery of neurons of diverse nature such as polysemantic, and superposition. Theoretically, only the ElasticNet regularization is capable of discovering neurons learning a singular function and multiple functions. The other methods such as Probeless are incapable of discovering multifunction neurons. None of the neuron interpretation methods explicitly identify superposition neurons. Explicit modeling of neurons of different nature in a neuron interpretation method may result in discovering novel sets of neurons. The evaluation methods including our proposed framework do not explicitly compare neurons of different types.

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

# 7 Appendix

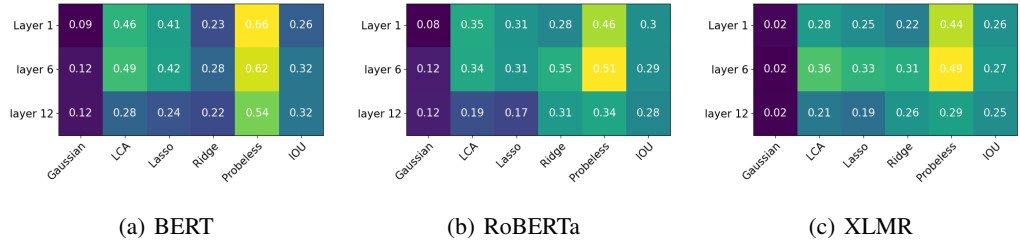

(a) BERT  (b) RoBERTa  (c) XLMR

Figure 2: *Task: POS,* Pairwise comparison of MeanSelect with other methods using 10–50 neurons

## 7.1 Case Study

**Mean Select:**  We propose a new corpus-based method, `MeanSelect`, for neuron ranking as a case-study to illustrate how researchers can use our proposed methodology. Kádár et al. (2017b) generated explanations for neurons by extracting top-N 5-gram context for each neuron based on the magnitude of their activations, followed by human annotation. Na et al. (2019) removed the human-in-the-loop by extracting concepts of various granularities from a parsed tree and aligned highly activating neurons to the concepts. Inspired by these works, we propose a novel method, `MeanSelect`, that generates a ranking of neurons with respect to a concept. The intuition is that a neuron learning a concept will have consistently high activations across different contexts where that concept appears. However, there may be a neuron that always activates with high value irrespective of the concept. A difference in the mean value of the neuron activating the concept and all other concepts will provide the true behavior of the neuron for the concept. Following our notation in Section 2, the score of a given neuron $n$ is defined as follows:

$$R(n, \mathcal{C}) = \frac{\mu(\mathcal{C}) - \mu(\hat{\mathcal{C}})}{n_{max} - n_{min}} \tag{10}$$

where $\mu(\mathcal{C})$ is the average, $n_{max}$ is the max and $n_{min}$ is the min of activations $z(n, w)$ where $w \in \mathcal{C}$, and $\mu(\hat{\mathcal{C}})$ is the average of activations over the random concept set.

**Compatibility Metrics**  Table 4 shows the compatibility score of `MeanSelect` using the representation of 3 different layers and compared it with the Random selection of neurons. The high compatibility scores show that `MeanSelect` discovers neurons that are endorsed by other neuron interpretation methods. This serves as a measure of confidence in the proposed method. Moreover, one may observe that the method has a relatively lower score for the last layer compared to other layers, giving insight into potential improvements that can be made to the behavior of this method.

**Pairwise Comparison**  The pairwise comparison of methods further provides insights into how the new method relates to other methods in terms of the resulting neurons. Figure 2 shows the heatmaps of three pre-trained models. MeanSelect has the highest overlap with the Probeless method and except Gaussian, it shows an overlap of at least 0.23 points with other methods. While the high overlap of MeanSelect with Probeless is not surprising given both are based on similar intuitions, the overlap with LCA and L1 shows that the method is selecting a diverse set of neurons captured by a variety of methods. Similar to the discussion on compatibility score, here we observe a substantial overlap drop in the last layer and this highlights the potential vulnerability of the method to certain representations.

## 7.2 Concept Datasets

We consider concepts from three linguistic tasks: parts of speech tags (POS, Marcus et al., 1993), semantic tags (SEM, Abzianidze et al., 2017) and syntactic chunking (Chunking) using CoNLL 2000 shared task dataset (Tjong Kim Sang & Buchholz, 2000). For the POS dataset, we used 20 concepts

Table 4: *Task: POS, Model: BERT,* Average `NeuronVote` score of MeanSelect using 10–50 neurons

| Layers | BERT | | | RoBERTa | | | XLMR | | |
|---|---|---|---|---|---|---|---|---|---|
| | 1 | 6 | 12 | 1 | 6 | 12 | 1 | 6 | 12 |
| Random | 0.019 | 0.021 | 0.023 | 0.020 | 0.019 | 0.017 | 0.019 | 0.017 | 0.022 |
| MeanSelect | 0.464 | 0.476 | 0.402 | 0.392 | 0.451 | 0.328 | 0.368 | 0.408 | 0.253 |

Table 5: *Task: Semantic tagging,* Average `NeuronVote` compatibility scores across Semantic tagging concepts when selecting the top 10, 30, and 50 neurons from layers 1, 6 and 12. Bold numbers, underline numbers, and dashed numbers show the first, second, and third best scores respectively

| Layers | BERT | | | RoBERTa | | | XLMR | | |
|---|---|---|---|---|---|---|---|---|---|
| | 1 | 6 | 12 | 1 | 6 | 12 | 1 | 6 | 12 |
| Random | 0.018 | 0.013 | 0.017 | 0.011 | 0.026 | 0.017 | 0.026 | 0.014 | 0.026 |
| Gaussian | 0.257 | 0.256 | 0.222 | 0.237 | 0.282 | 0.245 | 0.176 | 0.256 | 0.195 |
| LCA | **0.474** | **0.541** | 0.385 | 0.488 | 0.493 | 0.347 | 0.309 | 0.455 | 0.367 |
| Lasso | 0.407 | 0.492 | 0.322 | 0.391 | 0.460 | 0.328 | 0.294 | 0.396 | 0.358 |
| Ridge | 0.316 | 0.343 | 0.361 | 0.468 | 0.492 | 0.575 | 0.372 | 0.430 | **0.473** |
| Probeless | 0.450 | 0.501 | **0.476** | **0.547** | **0.586** | **0.640** | **0.495** | **0.571** | 0.464 |
| IoU | 0.344 | 0.332 | 0.380 | 0.288 | 0.287 | 0.242 | 0.277 | 0.262 | 0.282 |

which have a total dataset size of 40137. These concepts include VBG (777), VBZ (908), NNPS (204), DT (4015), TO (1177), CD (1935), JJ (2836), PRP (801), MD (463), RB (1348), VBP (534), VB (1244), NNS (3021), VBN (1082), POS (433), IN (5039), NN (6660), CC (1220), NNP (4698), and VBD (1742).

For the SEM dataset, we used three concepts which have a total dataset size of 120941. These concepts include IST (72240), NOW (24137) and EXS (24564). We used 10 concepts from Chunking which have a total dataset size of 220606. These concepts include B-ADJP (2493), B-ADVP (5081), B-NP (67285), B-PP (26005), B-VP (26078), I-ADJP (805), I-ADVP (532), I-NP (77368) I-PP (339) and I-VP (14620). For all the datasets used in the experiments, we use train/valid/test split 70%, 15% and 15%.

## 7.3 Results

### 7.3.1 Semantic Tagging Concepts

For the SEM dataset, we sample 20000 sentences for experimental validation with train/valid/test split 70%, 15% and 15%. We select three tags: IST (intersective), NOW (present tense) and EXS (untensed simple event). Table 5 presents the average `NeuronVote` scores across three models. We observed identical trends to that of POS i.e. Probeless is the most consistent method, LCA and Lasso are second best methods but they suffer on the last layers.

## 7.4 Chunking Concepts

Figures 3 and 4 show layer-wise results for the two voting methods proposed in the paper, across the three understudied models (BERT, RoBERTa and XLM-R). The results show that voting methods consistently rank the *Probeless* method as the most compatible in terms of neuron rankings across the layers. We are including detailed results in Tables 5–10 give detailed results with exact numbers.

Table 6: *Task: Chunking,* Average `NeuronVote` compatibility scores across Chunking concepts when selecting the top 10, 30, and 50 neurons from layers 1, 6 and 12. Bold numbers, underline numbers, and dashed numbers show the first, second, and third best scores respectively

| | BERT | | | RoBERTa | | | XLMR | | |
| --- | --- | --- | --- | --- | --- | --- | --- | --- | --- |
| Layers | 1 | 6 | 12 | 1 | 6 | 12 | 1 | 6 | 12 |
| Random | 0.018 | 0.017 | 0.023 | 0.022 | 0.018 | 0.022 | 0.023 | 0.019 | 0.014 |
| Gaussian | 0.122 | 0.181 | 0.174 | 0.111 | 0.163 | 0.164 | 0.110 | 0.121 | 0.143 |
| LCA | 0.395 | 0.469 | 0.300 | 0.440 | 0.422 | 0.328 | 0.336 | 0.447 | 0.388 |
| Lasso | 0.396 | 0.472 | 0.301 | 0.399 | 0.395 | 0.322 | 0.366 | 0.425 | 0.395 |
| Ridge | 0.235 | 0.255 | 0.256 | 0.303 | 0.330 | 0.386 | 0.254 | 0.289 | 0.410 |
| Probeless | **0.465** | **0.506** | **0.422** | **0.502** | **0.500** | **0.514** | **0.499** | **0.507** | **0.463** |
| IoU | 0.346 | 0.361 | 0.321 | 0.285 | 0.298 | 0.244 | 0.319 | 0.283 | 0.352 |

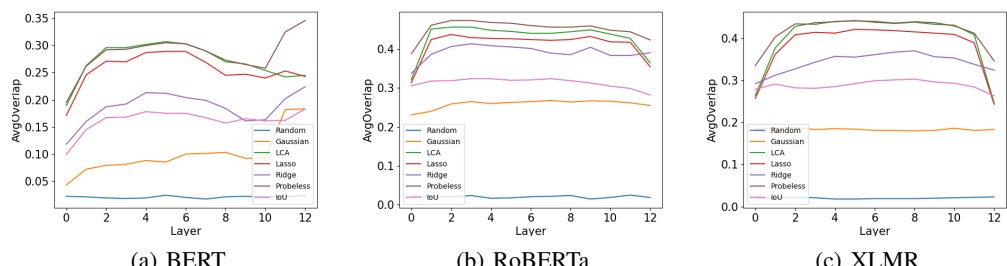

(a) BERT   (b) RoBERTa   (c) XLMR

Figure 3: `AvgOverlap` score across different Layers in different models. All methods perform better than the baseline, Random. Probeless is the most consistent method across all models, concepts and across all layers. It is among the top methods with LCA and Lasso. However, LCA and Lasso show low score on last layers. The performance of Gaussian deteriorates significantly and is closer to Random when applied on XLMR.

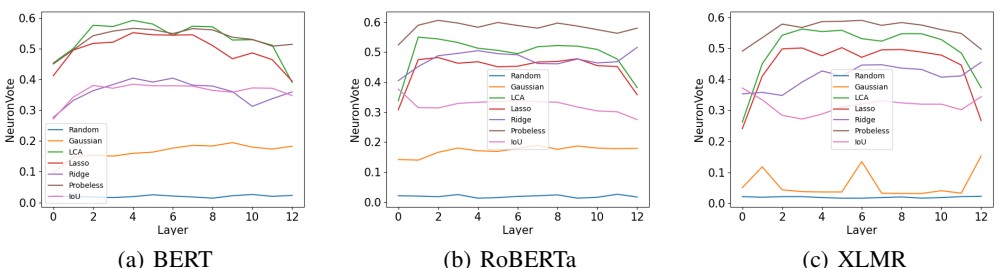

(a) BERT   (b) RoBERTa   (c) XLMR

Figure 4: `NeuronVote` score across different Layers in different models. All methods perform better than the baseline, Random. Probeless is the most consistent method across all models, concepts and across all layers. It is among the top methods with LCA and Lasso. However, LCA and Lasso show low score on last layers.

Table 7: This is an extension of Table.3. Average `AvgOverlap` compatibility scores across concepts when selecting the top 10, 30, and 50 neurons. Bold numbers show the best scores. Probeless, LCA are the top performing methods. However, Probeless is most consistent performing method. LCA drops substantially for the last layers.

| | BERT | | | | | | | | | | | | |
|---|---|---|---|---|---|---|---|---|---|---|---|---|---|
| Layers | 0 | 1 | 2 | 3 | 4 | 5 | 6 | 7 | 8 | 9 | 10 | 11 | 12 |
| Random | 0.022 | 0.021 | 0.019 | 0.018 | 0.019 | 0.024 | 0.020 | 0.017 | 0.021 | 0.022 | 0.021 | 0.022 | 0.023 |
| Gaussian | 0.043 | 0.072 | 0.079 | 0.081 | 0.088 | 0.085 | 0.100 | 0.101 | 0.103 | 0.092 | 0.093 | 0.182 | 0.183 |
| LCA | 0.189 | **0.263** | **0.296** | **0.296** | **0.302** | **0.307** | **0.303** | **0.290** | 0.27 | **0.266** | 0.254 | 0.242 | 0.245 |
| Lasso | 0.171 | 0.246 | 0.271 | 0.270 | 0.287 | 0.289 | 0.289 | 0.269 | 0.245 | 0.247 | 0.24 | 0.253 | 0.243 |
| Ridge | 0.118 | 0.16 | 0.187 | 0.192 | 0.213 | 0.212 | 0.204 | 0.199 | 0.184 | 0.161 | 0.163 | 0.202 | 0.224 |
| Probeless | **0.195** | 0.262 | 0.292 | 0.293 | 0.3 | 0.305 | **0.303** | 0.29 | **0.273** | 0.265 | **0.258** | **0.325** | **0.346** |
| IoU | 0.099 | 0.145 | 0.167 | 0.168 | 0.178 | 0.175 | 0.175 | 0.167 | 0.157 | 0.165 | 0.161 | 0.162 | 0.183 |

Table 8: This is an extension of Table.3. Average `NeuronVote` compatibility scores across concepts when selecting the top 10, 30, and 50 neurons.

| | BERT | | | | | | | | | | | | |
|---|---|---|---|---|---|---|---|---|---|---|---|---|---|
| Layers | 0 | 1 | 2 | 3 | 4 | 5 | 6 | 7 | 8 | 9 | 10 | 11 | 12 |
| Random | 0.022 | 0.019 | 0.018 | 0.016 | 0.019 | 0.025 | 0.021 | 0.018 | 0.014 | 0.022 | 0.026 | 0.020 | 0.023 |
| Gaussian | 0.097 | 0.147 | 0.154 | 0.151 | 0.160 | 0.164 | 0.177 | 0.186 | 0.184 | 0.195 | 0.180 | 0.174 | 0.183 |
| LCA | **0.454** | **0.501** | **0.577** | **0.573** | **0.593** | **0.581** | 0.544 | **0.574** | **0.572** | 0.529 | **0.530** | **0.512** | 0.391 |
| Lasso | 0.413 | 0.496 | 0.518 | 0.522 | 0.553 | 0.546 | 0.545 | 0.546 | 0.511 | 0.468 | 0.487 | 0.465 | 0.395 |
| Ridge | 0.277 | 0.332 | 0.364 | 0.384 | 0.405 | 0.392 | 0.405 | 0.382 | 0.379 | 0.361 | 0.313 | 0.338 | 0.360 |
| Probeless | 0.451 | 0.497 | 0.543 | 0.558 | 0.567 | 0.563 | **0.550** | 0.566 | 0.562 | **0.538** | **0.531** | 0.509 | 0.515 |
| IoU | 0.272 | 0.343 | 0.381 | 0.372 | 0.385 | 0.380 | 0.380 | 0.379 | 0.365 | 0.359 | 0.373 | 0.372 | 0.348 |

Table 9: This is an extension of Table.3. Average `AvgOverlap` compatibility scores across concepts when selecting the top 10, 30, and 50 neurons.

| | RoBERTa | | | | | | | | | | | | |
|---|---|---|---|---|---|---|---|---|---|---|---|---|---|
| Layers | 0 | 1 | 2 | 3 | 4 | 5 | 6 | 7 | 8 | 9 | 10 | 11 | 12 |
| Random | 0.021 | 0.020 | 0.019 | 0.023 | 0.016 | 0.017 | 0.020 | 0.021 | 0.023 | 0.014 | 0.018 | 0.024 | 0.018 |
| Gaussian | 0.231 | 0.240 | 0.259 | 0.265 | 0.260 | 0.263 | 0.265 | 0.268 | 0.264 | 0.267 | 0.266 | 0.262 | 0.255 |
| LCA | 0.322 | 0.452 | 0.457 | 0.457 | 0.449 | 0.446 | 0.441 | 0.441 | 0.445 | 0.450 | 0.439 | 0.428 | 0.365 |
| Lasso | 0.314 | 0.425 | 0.438 | 0.430 | 0.428 | 0.427 | 0.425 | 0.423 | 0.425 | 0.433 | 0.419 | 0.418 | 0.355 |
| Ridge | 0.338 | 0.387 | 0.407 | 0.414 | 0.409 | 0.406 | 0.402 | 0.390 | 0.386 | 0.405 | 0.384 | 0.384 | 0.391 |
| Probeless | **0.388** | **0.462** | **0.474** | **0.474** | **0.469** | **0.467** | **0.461** | **0.457** | **0.457** | **0.460** | **0.449** | **0.445** | **0.424** |
| IoU | 0.306 | 0.318 | 0.319 | 0.324 | 0.324 | 0.320 | 0.321 | 0.324 | 0.319 | 0.313 | 0.305 | 0.299 | 0.282 |

Table 10: This is an extension of Table.3. Average `NeuronVote` compatibility scores across concepts when selecting the top 10, 30, and 50 neurons.

| | RoBERTa | | | | | | | | | | | | |
|---|---|---|---|---|---|---|---|---|---|---|---|---|---|
| Layers | 0 | 1 | 2 | 3 | 4 | 5 | 6 | 7 | 8 | 9 | 10 | 11 | 12 |
| Random | 0.021 | 0.020 | 0.018 | 0.025 | 0.013 | 0.015 | 0.019 | 0.021 | 0.024 | 0.013 | 0.016 | 0.026 | 0.017 |
| Gaussian | 0.142 | 0.140 | 0.166 | 0.180 | 0.171 | 0.169 | 0.179 | 0.189 | 0.176 | 0.187 | 0.180 | 0.178 | 0.179 |
| LCA | 0.338 | 0.550 | 0.544 | 0.532 | 0.513 | 0.506 | 0.495 | 0.518 | 0.522 | 0.520 | 0.509 | 0.477 | 0.382 |
| Lasso | 0.308 | 0.475 | 0.482 | 0.463 | 0.468 | 0.451 | 0.453 | 0.467 | 0.469 | 0.478 | 0.455 | 0.452 | 0.358 |
| Ridge | 0.405 | 0.452 | 0.488 | 0.496 | 0.505 | 0.495 | 0.491 | 0.462 | 0.461 | 0.477 | 0.464 | 0.468 | 0.516 |
| Probeless | **0.524** | **0.590** | **0.606** | **0.597** | **0.583** | **0.599** | **0.589** | **0.580** | **0.597** | **0.587** | **0.575** | **0.563** | **0.580** |
| IoU | 0.377 | 0.315 | 0.314 | 0.329 | 0.333 | 0.335 | 0.338 | 0.335 | 0.333 | 0.317 | 0.304 | 0.301 | 0.275 |

Table 11: This is an extension of Table.3. Average `AvgOverlap` compatibility scores across concepts when selecting the top 10, 30, and 50 neurons.

| | XLMR | | | | | | | | | | | | |
|---|---|---|---|---|---|---|---|---|---|---|---|---|---|
| Layers | 0 | 1 | 2 | 3 | 4 | 5 | 6 | 7 | 8 | 9 | 10 | 11 | 12 |
| Random | 0.020 | 0.021 | 0.022 | 0.021 | 0.018 | 0.018 | 0.019 | 0.019 | 0.019 | 0.020 | 0.021 | 0.022 | 0.023 |
| Gaussian | 0.185 | 0.186 | 0.188 | 0.183 | 0.185 | 0.184 | 0.181 | 0.181 | 0.180 | 0.181 | 0.186 | 0.181 | 0.183 |
| LCA | 0.264 | 0.377 | 0.428 | **0.437** | **0.439** | **0.442** | 0.438 | **0.435** | **0.438** | 0.433 | **0.431** | 0.408 | 0.245 |
| Lasso | 0.257 | 0.362 | 0.408 | 0.414 | 0.412 | 0.421 | 0.420 | 0.418 | 0.415 | 0.412 | 0.409 | 0.389 | 0.243 |
| Ridge | 0.292 | 0.312 | 0.327 | 0.343 | 0.357 | 0.355 | 0.361 | 0.367 | 0.370 | 0.356 | 0.353 | 0.338 | 0.324 |
| Probeless | **0.335** | **0.403** | **0.434** | 0.433 | **0.440** | **0.441** | **0.440** | 0.436 | **0.439** | **0.437** | 0.429 | **0.412** | **0.346** |
| IoU | 0.279 | 0.291 | 0.282 | 0.281 | 0.285 | 0.292 | 0.299 | 0.301 | 0.303 | 0.296 | 0.293 | 0.284 | 0.263 |

Table 12: This is an extension of Table.3. Average `NeuronVote` compatibility scores across concepts when selecting the top 10, 30, and 50 neurons.

| Layers | | | | | | | XLMR | | | | | | |
|---|---|---|---|---|---|---|---|---|---|---|---|---|---|
| | 0 | 1 | 2 | 3 | 4 | 5 | 6 | 7 | 8 | 9 | 10 | 11 | 12 |
| Random | 0.022 | 0.020 | 0.022 | 0.022 | 0.019 | 0.017 | 0.017 | 0.019 | 0.021 | 0.017 | 0.019 | 0.022 | 0.023 |
| Gaussian | 0.051 | 0.118 | 0.044 | 0.038 | 0.037 | 0.037 | 0.135 | 0.033 | 0.032 | 0.032 | 0.041 | 0.033 | 0.153 |
| LCA | 0.262 | 0.450 | 0.542 | 0.562 | 0.554 | 0.558 | 0.531 | 0.523 | 0.547 | 0.547 | 0.528 | 0.485 | 0.373 |
| Lasso | 0.241 | 0.410 | 0.498 | 0.501 | 0.476 | 0.502 | 0.471 | 0.495 | 0.496 | 0.488 | 0.478 | 0.446 | 0.267 |
| Ridge | 0.353 | 0.358 | 0.348 | 0.391 | 0.427 | 0.411 | 0.446 | 0.447 | 0.436 | 0.432 | 0.407 | 0.411 | 0.455 |
| Probeless | **0.491** | **0.534** | **0.578** | **0.567** | **0.586** | **0.587** | **0.590** | **0.574** | **0.583** | **0.575** | **0.560** | **0.548** | **0.497** |
| IoU | 0.372 | 0.334 | 0.284 | 0.272 | 0.288 | 0.311 | 0.320 | 0.331 | 0.324 | 0.320 | 0.320 | 0.302 | 0.344 |

Table 13: This is an extension of Table.4. Average `AvgOverlap` score of MeanSelect when selecting 10, 30, and 50 neurons for all layers

| Layers | 0 | 1 | 2 | 3 | 4 | 5 | 6 | 7 | 8 | 9 | 10 | 11 | 12 |
|---|---|---|---|---|---|---|---|---|---|---|---|---|---|
| | | | | | | | BERT | | | | | | |
| Random | 0.022 | 0.021 | 0.019 | 0.018 | 0.019 | 0.022 | 0.020 | 0.017 | 0.021 | 0.022 | 0.021 | 0.022 | 0.023 |
| MeanSelect | 0.266 | 0.351 | 0.363 | 0.362 | 0.365 | 0.369 | 0.374 | 0.370 | 0.356 | 0.349 | 0.343 | 0.327 | 0.283 |
| | | | | | | | RoBERTa | | | | | | |
| Random | 0.021 | 0.020 | 0.019 | 0.023 | 0.016 | 0.017 | 0.020 | 0.021 | 0.023 | 0.014 | 0.018 | 0.024 | 0.018 |
| MeanSelect | 0.252 | 0.294 | 0.307 | 0.317 | 0.326 | 0.328 | 0.323 | 0.320 | 0.314 | 0.308 | 0.283 | 0.269 | 0.239 |
| | | | | | | | XLMR | | | | | | |
| Random | 0.020 | 0.021 | 0.022 | 0.021 | 0.018 | 0.018 | 0.019 | 0.019 | 0.019 | 0.020 | 0.021 | 0.022 | 0.023 |
| MeanSelect | 0.233 | 0.246 | 0.245 | 0.247 | 0.264 | 0.273 | 0.294 | 0.292 | 0.281 | 0.261 | 0.259 | 0.241 | 0.159 |

Table 14: This is an extension of Table.4. Average `NeuronVote` score of MeanSelect when selecting 10, 30, and 50 neurons for all layers

| Layers | 0 | 1 | 2 | 3 | 4 | 5 | 6 | 7 | 8 | 9 | 10 | 11 | 12 |
|---|---|---|---|---|---|---|---|---|---|---|---|---|---|
| | | | | | | | BERT | | | | | | |
| Random | 0.022 | 0.019 | 0.018 | 0.016 | 0.019 | 0.025 | 0.021 | 0.018 | 0.014 | 0.022 | 0.026 | 0.020 | 0.023 |
| MeanSelect | 0.361 | 0.476 | 0.462 | 0.458 | 0.470 | 0.468 | 0.479 | 0.488 | 0.486 | 0.463 | 0.457 | 0.432 | 0.400 |
| | | | | | | | RoBERTa | | | | | | |
| Random | 0.021 | 0.020 | 0.018 | 0.025 | 0.013 | 0.015 | 0.019 | 0.021 | 0.024 | 0.013 | 0.016 | 0.026 | 0.017 |
| MeanSelect | 0.358 | 0.388 | 0.395 | 0.403 | 0.426 | 0.423 | 0.455 | 0.438 | 0.437 | 0.406 | 0.383 | 0.374 | 0.326 |
| | | | | | | | XLMR | | | | | | |
| Random | 0.022 | 0.020 | 0.022 | 0.022 | 0.019 | 0.017 | 0.017 | 0.019 | 0.021 | 0.017 | 0.019 | 0.022 | 0.023 |
| MeanSelect | 0.376 | 0.357 | 0.356 | 0.342 | 0.364 | 0.392 | 0.414 | 0.416 | 0.400 | 0.372 | 0.381 | 0.360 | 0.243 |

Table 15: Comparison of NeuronVote score of Probeless and LCA under different number of consensus methods. In majority of the cases, Probeless outperformed LCA irrespective of the methods used for consensus. Moreover, in three cases where LCA is better than Probeless, their results are comparable.

| Num. Methods | Consensus Methods | Probeless | LCA |
|---|---|---|---|
| 1 | Gaussian | **0.110** | 0.096 |
| | IoU | **0.274** | 0.216 |
| | Lasso | 0.449 | **0.516** |
| | Ridge | **0.270** | 0.240 |
| 2 | Gaussian,IoU | **0.233** | 0.177 |
| | Gaussian,Ridge | **0.202** | 0.174 |
| | Gaussian,Lasso | 0.272 | **0.275** |
| | Lasso,IoU | **0.351** | 0.329 |
| | Lasso,Ridge | 0.404 | **0.410** |
| | Ridge,IoU | **0.292** | 0.243 |
| 3 | Gaussian,Ridge,IoU | **0.265** | 0.229 |
| | Gaussian,Lasso,Ridge | **0.337** | 0.327 |
| | Gaussian,Lasso,IoU | **0.327** | 0.305 |
| | Lasso,Ridge,IoU | **0.373** | 0.351 |
| 4 | Gaussian,Lasso,Ridge,IoU | **0.368** | 0.342 |

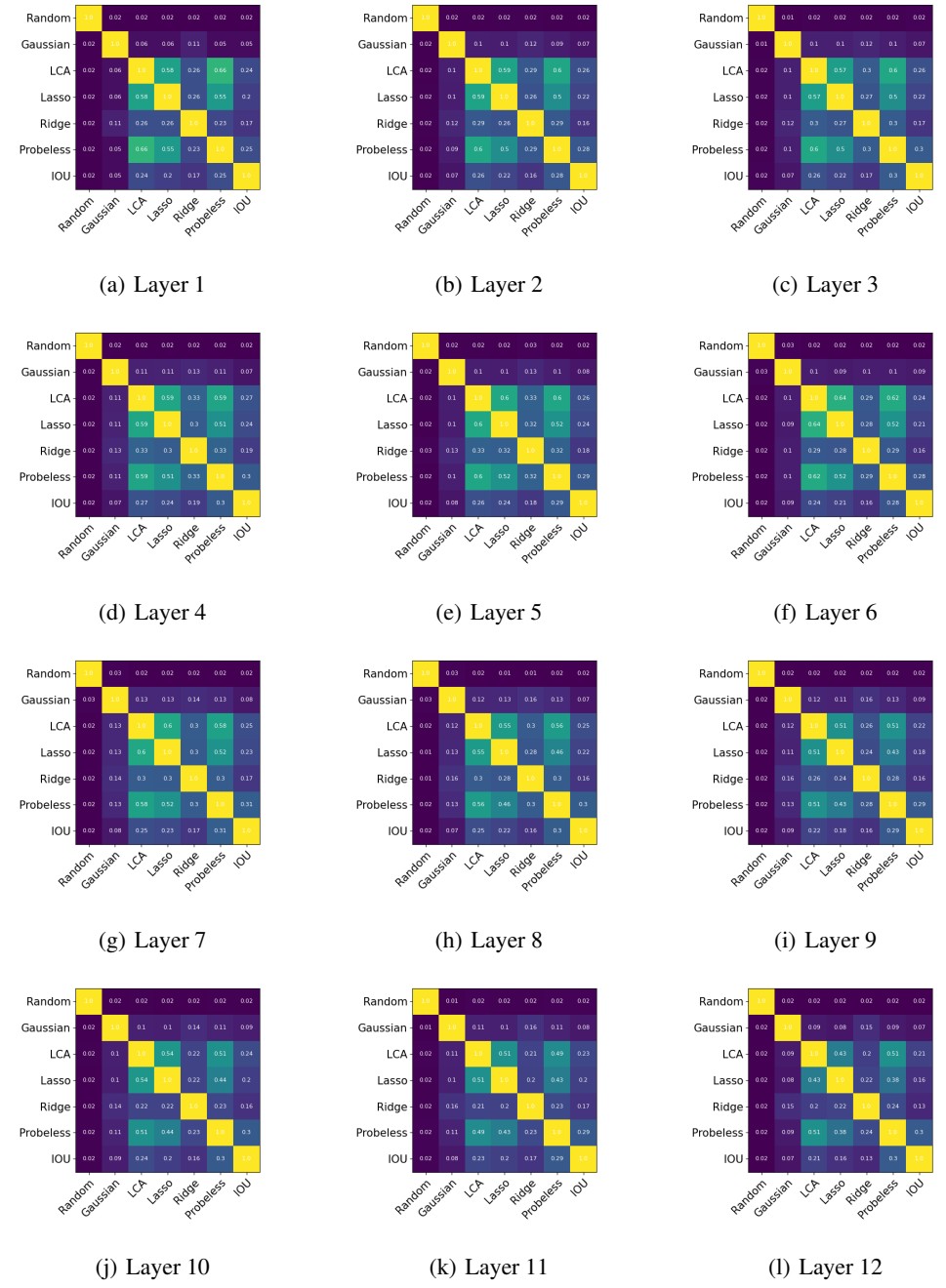

Figure 5: This is an extension of Figure.1. Comparing average overlap of top 10-50 neurons across methods for BERT

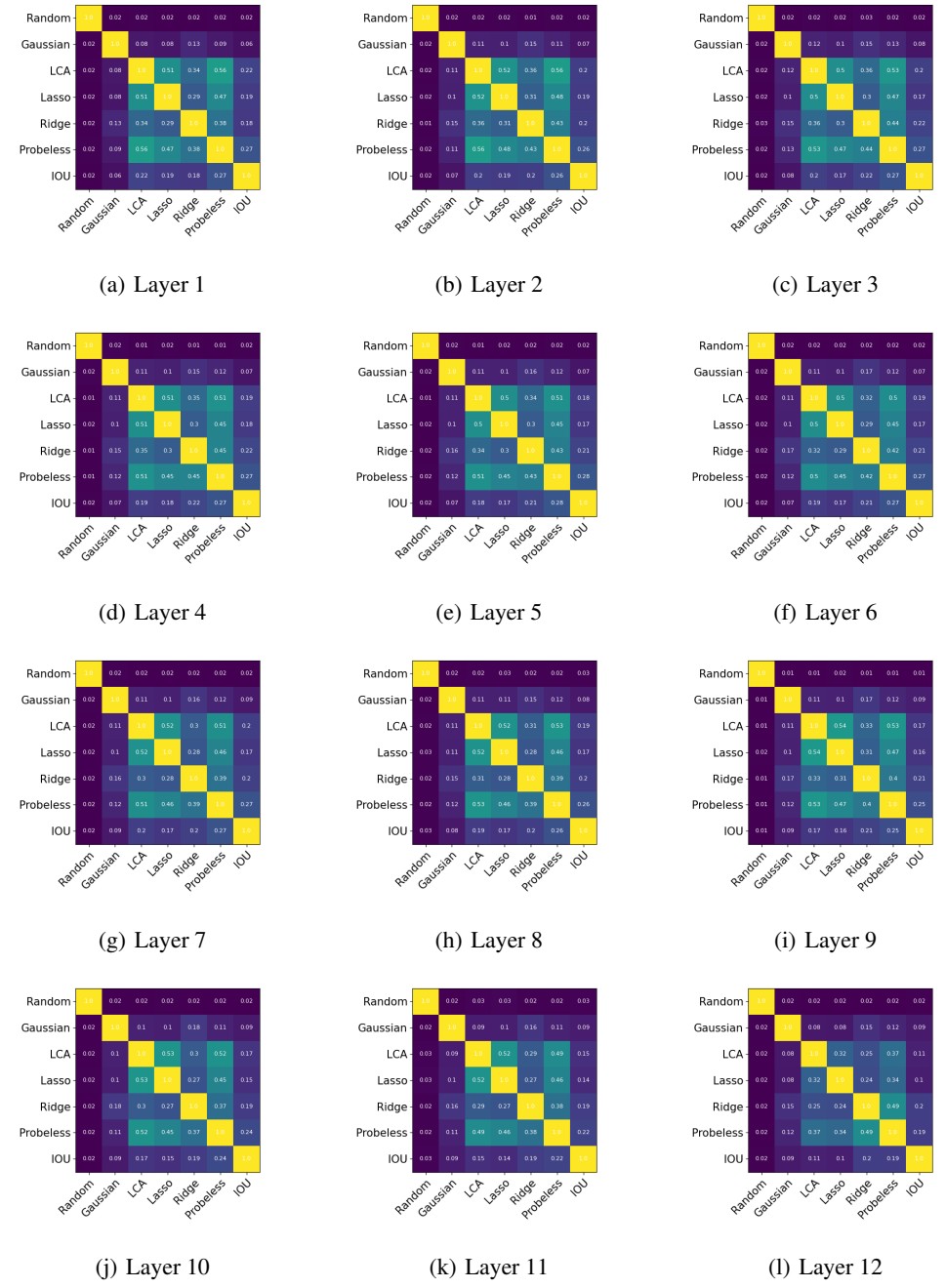

Figure 6: This is an extension of Figure.1. Comparing average overlap of top 10-50 neurons across methods for RoBERTa

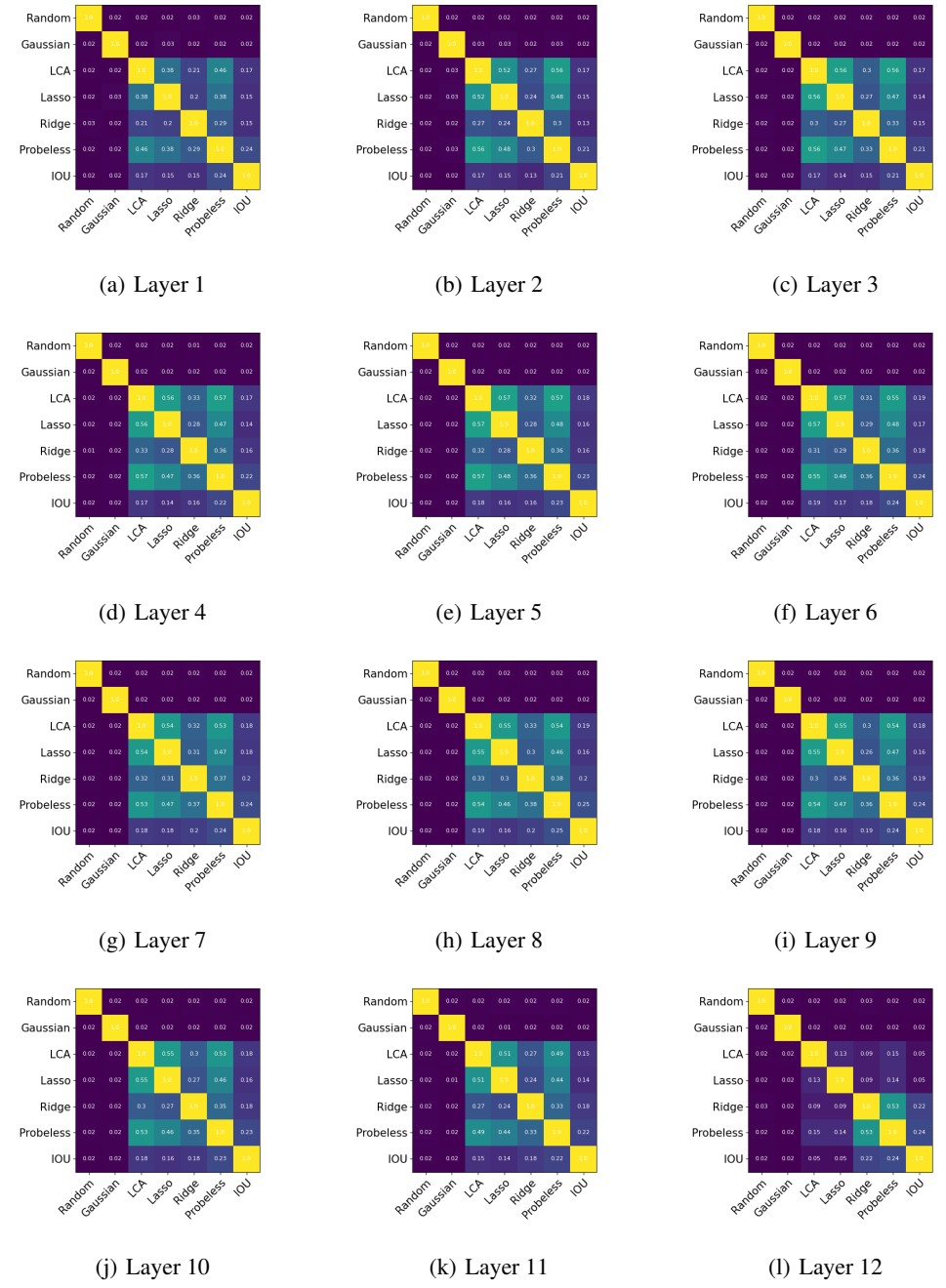

Figure 7: This is an extension of Figure.1. Comparing average overlap of top 10-50 neurons across methods for XLMR

Table 16: Comparison of NeuronVote score of Probeless and Lasso under different number of consensus methods. In all cases, Probeless shows better results irrespective of the methods used for consensus.

| Num of Consensus Methods | Consensus Methods | Probeless | Lasso |
|---|---|---|---|
| 1 | Gaussian | **0.110** | 0.092 |
| | IoU | **0.274** | 0.195 |
| | LCA | **0.524** | 0.516 |
| | Ridge | **0.270** | 0.237 |
| 2 | Gaussian,IoU | **0.233** | 0.159 |
| | Gaussian,LCA | **0.291** | 0.267 |
| | Gaussian,Ridge | **0.202** | 0.170 |
| | LCA,IoU | **0.357** | 0.300 |
| | LCA,Ridge | **0.387** | 0.329 |
| | Ridge,IoU | **0.292** | 0.227 |
| 3 | Gaussian,LCA,IoU | **0.346** | 0.289 |
| | Gaussian,LCA,Ridge | **0.356** | 0.321 |
| | Gaussian,Ridge,IoU | **0.265** | 0.214 |
| | LCA,Ridge,IoU | **0.424** | 0.396 |
| 4 | Gaussian,LCA,Ridge,IoU | **0.382** | 0.324 |

Table 17: Comparison of NeuronVote score of Probeless and IoU under different number of consensus methods. In all cases, Probeless shows better results irrespective of the methods used for consensus.

| Num of Consensus Methods | Consensus Methods | Probeless | IoU |
|---|---|---|---|
| 1 | Gaussian | **0.110** | 0.085 |
| | IoU | **0.449** | 0.195 |
| | Lasso-01 | **0.524** | 0.216 |
| | Ridge-01 | **0.270** | 0.160 |
| 2 | Gaussian,IoU | **0.272** | 0.155 |
| | Gaussian,Ridge-01 | **0.291** | 0.158 |
| | Gaussian,Lasso-01 | **0.202** | 0.136 |
| | Lasso-01,IoU | **0.404** | 0.206 |
| | Lasso-01,Ridge-01 | **0.498** | 0.209 |
| | Ridge-01,IoU | **0.424** | 0.217 |
| 3 | Gaussian,Ridge,IoU | **0.337** | 0.194 |
| | Gaussian,Lasso-01,Ridge-01 | **0.356** | 0.200 |
| | Gaussian,Lasso-01,IoU | **0.475** | 0.224 |
| | Lasso-01,Ridge-01,IoU | **0.434** | 0.206 |
| 4 | Gaussian,Lasso,Ridge-01,IoU | **0.426** | 0.221 |

