# OpenReview forum: "Evaluating Neuron Interpretation Methods of NLP Models"
_NeurIPS.cc/2023/Conference — NeurIPS 2023 poster_

### Official Review · Reviewer_F8Vs · 2023-07-05

**Soundness:** 2 fair
**Presentation:** 4 excellent
**Contribution:** 2 fair
**Rating:** 4
**Confidence:** 4

**Summary:**

This work investigates interpretation methods in NLP that identify which neurons in a neural network are most related to particular concepts (e.g. a specific part of speech). The key idea is to compare how consistent one method's ranking of neurons is (w.r.t. a specific concept) with that of all other considered methods. The assumption is that the method that produces the most consistent rankings is the one that is closest to the true ranking of important neurons. The authors consider a number of interpretation methods and pretrained language models and show that across several settings, the method that is most consistent is Probeless (Antverg and Belinkov, 2021).

**Strengths:**

- The work considers the problem of interpreting the neuron-level representations of neural networks, which seems to be a recently emerging subarea of interpretability within NLP. I'm not fully convinced by its utility but there are indeed several recent works in this area so perhaps there would be interest in this work
- The paper is mostly well written and motivated

**Weaknesses:**

The main weakness is the assumption that consensus with existing interpretability methods is desirable. This consensus of course depends on the other considered interpretability methods, and it's not clear that the number of total interpretability methods (6) is sufficient to lead to reliable results that are based on consensus.

**Questions:**

How do the insights about which interpretability methods is the most consistent change according to the considered interpretability methods? For example, can the authors remove 1,..n-1 interpretability methods from the consensus calculation and compare the results with those from the full set of n interpretability methods?

Minor questions / notes:

L100: are the examples supposed to be country names? The provided examples are cities.

There are a lot more "random"concepts in a corpus than target concepts. Are the random concepts that are used for evaluation sampled such that they are equal in number to the target concepts? How consistent is this evaluation over different random samples?

L185: why does the performance of a probe using random 100 neurons being higher than other neurons mean that the probe is memorizing? As the authors point out earlier in the manuscript, the concept knowledge is distributed so it makes sense that as one increases the number of random neurons that are being considered, the prediction performance would improve. If the authors were looking at randomly initialized models rather than pretrained models, then I would agree with their conclusion, but in the current setup, there seem to be alternate explanations for their results.


**Limitations:**

See under Weaknesses

---

> ### Author Rebuttal · Authors · 2023-08-09
>
> **R: …his consensus of course depends on the other considered interpretability methods, and it's not clear that the number of total interpretability methods (6) is sufficient to lead to reliable results…**
>
> A: We acknowledge the concern. We discussed the limitation of our approach in detail in Section 5. We would also like to emphasize the significance and critical nature of the problem we are addressing. Currently, there is no standardized metric available for comparing and evaluating neuron interpretation methods. This lack of standardization has led to scattered efforts in proposing new interpretation methods. Our work aims to address this gap by establishing a foundation for evaluating and comparing neuron interpretation methods, which will be continuously updated to include new interpretation methods.
> Regarding the choice of six interpretability methods: we did not explicitly reject an interpretation method from inclusion into our framework. The choice of six interpretability methods is based on the recently published survey on neuron interpretation (link: https://direct.mit.edu/tacl/article/doi/10.1162/tacl_a_00519/113852/Neuron-level-Interpretation-of-Deep-NLP-Models-A). Additionally, the six methods utilized in our paper were chosen for their theoretical diversity, resulting in a diverse array of discovered neurons.
> We welcome any discussion on alternative methods that the reviewer believes could help address the limitations of evaluating neuron interpretation techniques. We are open to considering and incorporating such suggestions to further enhance the robustness of our approach.
>
> **R: For example, can the authors remove 1,..n-1 interpretability methods from the consensus calculation and compare the results with those from the full set of n interpretability methods?**
>
> A: Thank you for suggesting an insightful experiment. In the paper, we considered a leave-one-out strategy to calculate compatibility of each method so for a set of N methods, we considered one method as a test method and used the rest N-1 methods to serve as the database of sets of discovered neurons. What you have suggested is opposite to this. We tried it for the rebuttal. We considered a combination of one, two, three and four methods to serve as a database of the sets of neurons and calculate the compatibility scores of the other methods. The trends in Table 1, 2 and 3 showed consistent results to that reported in the paper. We observe that only the inclusion of Lasso in the database, when there is only one or two methods, provides an edge to LCA during evaluation. However, this effect is minimized with the inclusion of more methods (greater than 2) in the consensus database. Rest, we did not find any correlation between the presence of a method in the database with the high compatibility score of a test method.
>
>
> **R: L100: are the examples supposed to be country names? The provided examples are cities.**
>
> A: Thank you for pointing out the typo. We will fix it to be city names.
>
>
> **R: Are the random concepts that are used for evaluation sampled such that they are equal in number to the target concepts?**
>
> A: Yes, we randomly selected “random” concepts equal in number to the target concept. We ran three random selection runs and found the results to be consistent.
>
>
> **R: L185: why does the performance of a probe using random 100 neurons being higher than other neurons mean that the probe is memorizing?**
>
> A: Thank you for pointing out the error. It is indeed due to the distributiveness of knowledge of the concept. Durrani et al. (2020) presented an empirical evidence of this using the controlled task. We have corrected it in the paper.

---

> > ### Comment · Reviewer_F8Vs · 2023-08-14
> >
> > Thanks for the response and the added experiment that is aimed to examine the robustness of the proposed consensus evaluation metrics to the differences in included approaches. Based on Tables 1,2, and 3 in the rebuttal PDF, it looks like there are different results when evaluating the same method (e.g. Probeless) against the same set of methods (e.g. Gaussian + LCA, or Gaussian + Ridge). I'm assuming this is a mistake. Please provide the corrected numbers.
> >
> > Abstracting away from the exact numbers, it looks like 3 of the evaluated methods -- Probeless, LCA, and Lasso -- agree more with each other than on average across all datasets. This is important because including two of these in the consensus methods and the 3rd as the method to be evaluated will bias the results towards this 3rd method over the remaining methods that are less similar (Ridge, Gaussian, IoU). Perhaps this a feature of these methods recovering something closer to the "true" set of important neurons, but that is only a speculation. This needs to be discussed.

---

> > > ### Author Response · Authors · 2023-08-15
> > >
> > > Thank you for your response.
> > >
> > > **re: error in table**
> > >
> > >
> > > We sincerely apologize for the error. Table 2 is accurate. Some rows are Table 1 are mislabelled (but the numbers remain the same) and in Table 3, the Probeless numbers for "LCA, Ridge" and "LCA, Ridge, IoU" were flipped.
> > >
> > >
> > > The final Table 1 is as follows:
> > >
> > >
> > > | Consensus Methods           | Probeless |  IoU  |
> > > |-----------------------------|:---------:|:-----:|
> > > | Gaussian                    |   0.110   | 0.085 |
> > > | Lasso                       |   0.449   | 0.195 |
> > > | LCA                         |   0.524   | 0.216 |
> > > | Ridge                       |   0.270   | 0.160 |
> > > | Gaussian, Lasso             |   0.272   | 0.155 |
> > > | Gaussian, LCA               |   0.291   | 0.158 |
> > > | Gaussian, Ridge             |   0.202   | 0.136 |
> > > | Lasso, Ridge                |   0.404   | 0.206 |
> > > | Lasso, LCA                  |   0.498   | 0.209 |
> > > | Ridge, LCA                  |   0.424   | 0.217 |
> > > | Gaussian, Ridge, Lasso      |   0.337   | 0.194 |
> > > | Gaussian, LCA, Ridge        |   0.356   | 0.200 |
> > > | Ridge, Lasso, LCA           |   0.475   | 0.224 |
> > > | Lasso, Gaussian, LCA        |   0.434   | 0.206 |
> > > | Gaussian, Lasso, Ridge, LCA |   0.426   | 0.221 |
> > >
> > >
> > > **Specifically:**
> > >
> > >
> > > In single method consensus, LCA and Lasso needs to be interchanged so row2 -> Lasso, row3 -> LCA
> > >
> > >
> > > In two methods consensus, the order of combinations is:
> > >
> > >
> > > Gaussian, Lasso
> > >
> > >
> > > Gaussian, LCA
> > >
> > >
> > > Gaussian, Ridge
> > >
> > >
> > > Lasso, Ridge
> > >
> > >
> > > Lasso, LCA
> > >
> > >
> > > Ridge, LCA
> > >
> > >
> > > Lastly, in three methods consensus, the order of combinations is:
> > >
> > >
> > > Gaussian, Ridge, Lasso
> > >
> > >
> > > Gaussian, LCA, Ridge
> > >
> > >
> > > Ridge, Lasso, LCA
> > >
> > >
> > > Lasso, Gaussian, LCA
> > >
> > >
> > > The trend did not change due to mislabelling. As you have observed, Probeless, LCA and Lasso have the highest overlap in terms of the discovered neurons.
> > >
> > >
> > >
> > > ## **re: Bias in consensus methods**
> > >
> > >
> > > Your observation is correct that Probeless, Lasso and LCA discovered the most similar top neurons, and this trend is also visible in the pairwise comparison provided in Figure 1 in the paper. We have discussed the point of potential bias in consensus in the limitation section. However, in this particular case, the similarity of the discovered neurons by these methods is less attributable to biases in the underlying methods, because they belong to two distinct theoretical classes of methods (classifier vs corpus-based). In other words, the high overlap among their discovered neurons is not due to methodological similarities, and hints that the behavior is observed because “these methods recovering something closer to the “true” set of important neurons.” However, this does not mean that these overlapping discovered neurons form a superset of “true” neurons with respect to the concept. There can be other true neurons that are not part of the overlapping set of neurons. One argument to support this point is to see the percentage of overlap between these methods (see Figure 1 in the paper). Despite high overlap, there are 30-40% neurons which are different among LCA, Probeless and Lasso. In any case, this is indeed a very insightful discussion in comparing neuron interpretation methods, and we will add it in the paper describing this spectrum of bias vs "true set" of neurons.

---

> > > > ### Author Response · Authors · 2023-08-20
> > > >
> > > > Dear reviewer, thank you for your time and valuable comments. We would be happy to discuss the rebuttal further and address any more questions/points of confusion.

---

### Official Review · Reviewer_x9Fa · 2023-07-06

**Soundness:** 3 good
**Presentation:** 3 good
**Contribution:** 2 fair
**Rating:** 4
**Confidence:** 3

**Summary:**

This work evaluates six different interpretation methods from a unified perspective. The authors focus on two challenges in this field: the absence of standard metrics and the lack of benchmarks. They propose two voting-based metrics to evaluate the compatibility among these six methods. Probeless consistently achieves the highest compatibility across all models based on their evaluation methodology. Further analytics experiments provide insight to the research community.

**Strengths:**

1. The findings of the interpretation methods mentioned in this paper may foster research on the topic.

2. The idea of this paper is slightly novel.

**Weaknesses:**

1. The selected six methods need to be more novel. More recent advancements in this field may exist rather than relying on L1 & L2 regularization.

2. The validation of the superiority of Probeless needs to be more comprehensive.

3. The mentioned problems (metrics and benchmarks) were not effectively addressed, making this method difficult to be reproduced in future applications.

**Questions:**

1. Are there additional performance comparison experiments to practically demonstrate the effectiveness of this method? For example, try to deactivate the most important 10% of neurons calculated by each interpretation method, and then observe whether Probeless exhibits the most significant performance decline.

2. To assess the efficacy of an interpretation method across various fields, is it essential to replicate multiple interpretation methods within those fields before applying the voting metrics? If so, this process can be quite challenging and should be acknowledged as a limitation in this work.

**Limitations:**

There is no potential negative societal impact.

---

> ### Author Rebuttal · Authors · 2023-08-09
>
> **R: The selected six methods need to be more novel. More recent advancements in this field may exist rather than relying on L1 & L2 regularization.**
>
> A: We selected the widely used and established neuron interpretation methods for NLP models, drawing from the existing literature. Please see the recent published survey on neuron interpretation
> (https://direct.mit.edu/tacl/article/doi/10.1162/tacl_a_00519/113852/Neuron-level-Interpretation-of-Deep-NLP-Models-A). Our selection of neuron interpretation methods is based on it.
> However, kindly note that the crux of our contribution lies not in the selection of these interpretation methods but rather in the introduction of a novel evaluation framework. This framework is specifically designed to streamline the assessment of new interpretation methods and facilitate meaningful comparisons of their results. By offering this evaluation tool, we aim to foster advancements in the field of neuron interpretation of NLP models, and encourage further research in this important area.
>
> **R: The validation of the superiority of Probeless needs to be more comprehensive.**
>
> A: We conducted a large set of experiments using three pre-trained models, across all layers, and using 50 concepts of diverse linguistic annotations such as parts of speech tagging, chunking and semantic tagging. This makes approximately 2000 settings in total. All of our results showed Probeless to be the most consistent method. LCA is another competitive method to Probeless with the exception of last layer representations. Appendix Tables 6 and 7 present the results using semantic tagging and chunking tasks. We included the set of experiments that you have proposed and we would be happy to consider and discuss if there are any suggestions of more experiments.
>
> **R: The mentioned problems (metrics and benchmarks) were not effectively addressed**
>
> A: We kindly seek clarification on the matter. We have integrated all methods into a single codebase, shared the evaluation metric code, and provided discovered neurons associated with various concepts to ensure the  reproducibility of the results. We are confident that replicating our findings for future applications will be a straightforward process.
>
> **R: Q1: …additional experiments…  For example, try to deactivate the most important 10% of neurons calculated by each interpretation method, and then observe whether Probeless exhibits the most significant performance decline.**
>
> A: Thanks for proposing an interesting experiment. As suggested, we compared the compatibility score by iteratively removing top N neurons from the ranking of each method and calculated the compatibility score for the next top neurons. Figure 1 in the PDF file showed the average results across layer 1, 6 and 12. Probeless maintains the top or competitive to the top compatibility score for all three models, BERT, RoBERTa and XLMR. This shows that the top neurons discovered by Probeless are consistently better.
>
> **R: Q2 To assess the efficacy of an interpretation method across various fields, is it essential….**
>
> A: This is correct. The compatibility metric relies on the availability of several methods targeting a common goal. The metric is of significant value when gold standard annotations are not available and are harder to make. We have added the following text in the limitation section to acknowledge this limitation.
>
> ```
> While the proposed framework is agnostic to methods used to produce ranking, in order to adapt it to other fields, it requires the presence of various methods targeting an identical goal.
> ```

---

> > ### Author Response · Authors · 2023-08-20
> >
> > Dear reviewer, thank you for your time and valuable comments. We would be happy to discuss the rebuttal and address other questions/points of confusion you may have.

---

### Official Review · Reviewer_QcFR · 2023-07-06

**Soundness:** 3 good
**Presentation:** 2 fair
**Contribution:** 3 good
**Rating:** 5
**Confidence:** 3

**Summary:**

This paper provides a comparative analysis of six neuron interpretation methods utilizing diverse concepts across three distinct pre-trained models and introduces an evaluation framework predicated on voting theory. Importantly, it offers the first comprehensive examination of multiple neuron interpretation methods and strives to mitigate the challenges in this field.  The authors note similarities among the most proficient techniques within layers of neurons, with these resemblances remaining regardless of methodological deviations. It is thereby suggested, existing neuronal interpretation methodologies might have focused on a reciprocal group of top-performing neurons.

The paper also discusses the lack of no recognised evaluative metrics along with the lack of gold annotations by suggesting an evaluation strategy that consists of two compatible metrics with a pairwise comparison. This approach could facilitate the creation of a means to evaluate new neuron interpretation methods.

**Strengths:**

The major strength of the paper lies in its originality; being the first work to create a ranking metric using voting theory for comparing the various neuron interpretation methods available. This methos should allow for autumatic comparison between existing and new neuron actiavtion methos. Furthermore, it embarked on a comprehensive comparative analysis via this new evaluation framework

**Weaknesses:**

The primary weakness of the paper lies in its disregard for significant, influential studies previously published in the domain, thus lacking a comprehensive deliberation on an extensive range of neuron activation methods studied by the mechanistic interpretability community. In addition, the paper grievously overlooks the complexities associated with polysemantic, superposition and neuroplastic neuron behaviors encountered in neuron activation.

Work worth considering are:

1- Elhage, et al - https://transformer-circuits.pub/2021/framework/index.html

2 - Olsson et al - https://transformer-circuits.pub/2022/in-context-learning-and-induction-heads/index.html

3- Elhage, et al - https://transformer-circuits.pub/2022/in-context-learning-and-induction-heads/index.html

4- Henighan et al - https://transformer-circuits.pub/2023/toy-double-descent/index.html

5 - Foote et al - https://arxiv.org/pdf/2305.19911.pdf

6 - Bills et al - https://openaipublic.blob.core.windows.net/neuron-explainer/paper/index.html

7 - wang et al - arXiv preprint arXiv:2211.00593

8 - Chan et al - https://www.alignmentforum.org/posts/JvZhhzycHu2Yd57RN/causal-scrubbing-a-method-for-rigorously-testing

9 - Tenney et al - https://arxiv.org/pdf/2008.05122.pdf


**Questions:**

Do I understand it right that if the methods under study are sub-optimal, the voting results can also be sub-optimal? In other words, the voting method does not provide any indications of how well a method may perform. Rather, it simply allows for a comparison between them.

Considering that we are in the early phases of understanding the black-box nature of ML models, is it appropriate to start discussing benchmarks now? I have my reservations due to our lack of comprehensive hypotheses regarding their inner mechanisms. Hence, it presents a dilemma because it's as if we might not be in the position to start comparing or benchmarking.

**Limitations:**

The authors provide some insightful limitations of thier work.

---

> ### Author Rebuttal · Authors · 2023-08-09
>
> **R: re: missing references**
>
> A: Thank you for pointing out the missing references. We are certainly open to including a broader view of the interpretation field in the paper to enhance its scope. We acknowledge that some relevant references on neuron interpretation, such as Foote et al and Bills et al, appeared on arxiv in May 2023, and unfortunately, we couldn't include them in the paper at the time of submission. We have added the related work in the paper that provides a broad overview of the interpretation field and defines the scope of our work (included in the overall rebuttal).
>
> **R: …the paper grievously overlooks the complexities associated with polysemantic, superposition and neuroplastic neuron behaviors encountered in neuron activation.**
>
>
> A: Thank you for raising this point. It is indeed worth noting that the neuron interpretation studies in NLP, that come under the scope of this paper, have this limitation that they do not adequately consider polysemantic, superposition and neuroplastic neuron behaviors. Among the current methods, only ElasticNet regularization holds the theoretical expectation of identifying polysemantic neurons, and a few works have provided empirical evidence to support this claim. However, a detailed analysis in this particular line of work is currently lacking. We plan to address this in the limitations section to provide a comprehensive perspective on the research landscape. Thank you again for the invaluable feedback in refining the paper's clarity and purpose.
> Following is the text added to the paper to bring attention to the limitation of current neuron interpretation methods.
>
>
> ```
> A limitation of current neuron interpretation methods is that they do not explicitly target the discovery of neurons of diverse nature such as polysemantic, and superposition. Theoretically, ElasticNet regularization is capable of discovering neurons learning a singular function and multiple functions. The other methods such as Probeless are incapable of discovering multifunction neurons. An explicit modeling of neurons of different nature in a neuron interpretation method may result in discovering novel sets of neurons.
> ```
>
> **R: Do I understand it right that if the methods under study are sub-optimal, the voting results can also be sub-optimal?**
>
> A: Given the lack of an objective evaluation criteria or gold annotations, we believe that the next best thing is a consensus based method like the one we have proposed. Iif all the methods employed are sub-optimal, the overall voting outcome will also be sub-optimal. However, we have carefully selected a range of well-established and methodologically diverse neuron interpretation methods as well as concepts for our study, to have a variety of discovered neurons for comparison, and provide the best evaluation currently possible. Given a new interpretation method, our setup provides its evaluation from the perspective of other neurons discovered by several other methods. This is the case with the Gaussian method presented in the paper where its ranking score is quite low but better than Random selection of neurons and it encourages the inventors to perform further evaluation of their method. The conclusion and limitation section (Section 5) discussed this limitation in detail.
>
> **R: …early phases of understanding the black-box nature of ML models, is it appropriate to start discussing benchmarks now…**
>
> A: We respect the opinion of the reviewer.  Interpretability is a very wide field with various subfields and every subfield is at a different level of maturity. For example, there are hundreds of papers within the last five years on representation analysis while the work on mechanistic interpretability is in its early stages. Neuron Interpretation (that our work is aiming at) has seen a number of methods in the last five years, but any comparison between these has been cursory because of a lack of a benchmark. Without any comparison, it is difficult to assess the progress of neuron analysis methods, understand their limitations, and guide further research endeavors.We believe that a benchmark like this is a first step towards a standardized yardstick to move forward as a subfield.

---

> > ### Comment · Reviewer_QcFR · 2023-08-14
> > **Official comment by Reviewer QcFR**
> >
> > I thank the authors for their responses. While I agree that Bills et al. did not appear until May 2023, Foote et al. was available on OpenReview at an ICLR workshop and Arxiv in April 2023. The suggested missing references do not encapsulate a broader view of interpretability but rather those closest to the approach you considered in your paper.
> >
> > I would encourage the authors to examine recent developments in neuron interpretation methods that address the issue of superposition. For example:
> > Sharkey et al.: https://www.alignmentforum.org/posts/z6QQJbtpkEAX3Aojj/interim-research-report-taking-features-out-of-superposition and Sharkey, Lee: https://arxiv.org/pdf/2305.03452.pdf
> > To strengthen the limitations section, I believe a more thorough discussion of the challenge of superposition is needed.
> >
> > I am unsatisfied with the answer in the last response regarding "...appropriate time to start discussing benchmarks...". I believe the works considered in this paper lie under the subfield of mechanistic interpretability and not in the broader interpretability field. If the authors believe otherwise, I would be interested in seeing a paragraph in the paper discussing how methods considered in this paper do not lie under mechanistic interpretability. Without such a discussion, I am less confident in recommending acceptance.

---

> > > ### Author Response · Authors · 2023-08-15
> > >
> > > Thank you for your response.
> > >
> > > ## **re: Foote et al**
> > >
> > > We are sorry for missing Foote et al. and we have added these references to the paper. In our general response to the reviewers, we provided a draft of the related work with the suggested references. We will improve it further for the final version of the paper.
> > >
> > >
> > > ## **re: superposition of neurons**
> > >
> > > Thank you for pointing out an exciting work on analyzing superposition neurons. Indeed most of the neuron interpretation with respect to a concept lacks in explicitly analyzing superposition neurons. We will add a discussion on the neurons of different nature and what the possible challenges are in discovering and evaluating them.
> > >
> > >
> > > ## **re: Mechanistic interpretability**
> > >
> > > Representation and Neuron analysis (scope of this paper) primarily involves examining the learned representations or embeddings within a neural network, aiming to unveil patterns and relationships within the data that the network has learned to capture [1,2], while Mechanistic interpretability takes a different route for model understanding and tries to reverse engineer the model itself [3]. While we make this distinction, we also understand that the area of interpretability is fairly new and it has seen major growth in recent years, and the lines between various subfields are blurrier and evolving with time. Moreover, perhaps neuron analysis itself can be divided into separate subfields, one targeting what knowledge is learned (methods in this paper), and one focusing more on the inner workings and how neurons interact with each other (closer to the mechanistic work). In this paper, our intention was to have a focus on neuron interpretation w.r.t knowledge learned specifically, and stick to the recent papers and survey on neuron interpretation to define our scope. We will expand this into a discussion in the paper to shed light on these subtleties and clarify the scope further.
> > >
> > > [1] Intrinsic Probing through Dimension Selection  https://aclanthology.org/2020.emnlp-main.15.pdf
> > >
> > > [2] Neuron-level Interpretation of Deep NLP Models: A Survey https://direct.mit.edu/tacl/article/doi/10.1162/tacl_a_00519/113852/Neuron-level-Interpretation-of-Deep-NLP-Models-A
> > >
> > > [3] https://transformer-circuits.pub/2022/mech-interp-essay/index.html

---

> > > > ### Comment · Reviewer_QcFR · 2023-08-15
> > > >
> > > > I thank the authors for incorporating my feedback on the related works and limitations sections. I will keep my score for this submission.

---

### Official Review · Reviewer_z745 · 2023-07-06

**Soundness:** 4 excellent
**Presentation:** 4 excellent
**Contribution:** 3 good
**Rating:** 7
**Confidence:** 4

**Summary:**

This paper proposes a standardized evaluation metric and benchmark for comparing various neuron interpretation methods, based on ideas from majority voting. The benchmark is based on the hypothesis that "neurons that are commonly discovered by different interpretation methods are more informative than others", and uses this hypothesis to rank & score methods by their alignment with the majority (in terms of the neurons they assign behaviors to). I discuss some thoughts/limitations of this hypothesis below (and the authors discuss it as well in their limitations section). The authors demonstrate empirically why the current commonly-used evaluation metric (fitting a classifier to the neurons to predict the concept) favors certain methods over others and suffers from the same common issues as probing classifiers in general. Then then propose two "compatibility"-based metrics, one set-based and one rank-based, for comparing the top-k neurons deemed informative for explaining a concept by each method, and provide in-depth analysis using the metrics to compare 6 different existing neuron interpretation methods (3 corpus-based, and 3 classifier-based).

Overall, I believe this paper is sound, well-executed, thorough, and clear, and will serve a useful role in the neural model interpretability sub-community. Based on the authors' response to my below questions, I may be inclined to raise my score further.

Edit: I read the authors' rebuttal and it addressed my concerns. I did not change my score as I originally insinuated above, in part due to finding validity in the other reviewers' critiques. I still think the paper should be accepted.

**Strengths:**

Originality & Significance:
- The paper serves an important role in the mechanistic interpretability sub-community, and is likely to be of good value to this community.
- I am not aware of any other attempts to organize methods in this manner, or propose a standardized benchmark.

Quality & Clarity:
- The experimental section is sound. The lack of ground-truth in makes evaluating explanation evaluation metric "faithfulness" difficult, but the authors have nonetheless provided a convincing set of results.
- The paper is well-organized and well-written. The paper uses proper mathematical notation throughout.
- The authors have promised to release the code upon publication.

**Weaknesses:**

Agreement with hypothesis underlying the benchmark:
- I am not sure I am totally convinced by the claim that the most informative neurons will be discovered by the most methods, given that neurons that are easily discoverable can often be those which are perhaps more simplistic in what they encode (e.g., only serve a singular function rather than multiple, concept is encoded only in that singular neuron rather than spread over multiple, etc.) However, the granularity and assumptions underlying neuron interpretation methods is a nascent research question in the interpretability community overall, so I will not fault the authors for their hypothesis (i.e., I think the paper is appropriately scoped and the hypothesis is reasonable in the context of existing/popular neuron interpretation methods). It would be nice for the authors to discuss limitations of the hypothesis (or neuron interpretation methods in general) in the limitations section.

More minor:
- The paper is lacking a related works section. While the methods studied in the paper are described in detail in Section 2, it would be nice to have some sort of summary paragraph of the field of neuron-level interpretation as a whole after the introduction to situate the work (and potentially mention other methods that are not tested in the paper).
- There is some minor conflation it seems between the decision to treat each neuron interpretation method as producing a ranked list vs. performing binary set membership classification. This is particularly confusing with the classifier methods, which are directly trained to do the latter (4.1) but tested on the former via the NeuronVote method. However, the inclusion of the set-based metric, AvgOverlap, mitigates this issue to some extent. I think it would be good for the authors to clarify early in the paper that each method *can* be viewed as producing a ranking, even though this is not always how the methods have been proposed or used in practice, and that the inclusion/comparison of *both* evaluation metrics is designed to provide a view that does not unnecessarily favor one method over the other, by considering both the set membership and the rank views.

**Questions:**

- L12: colon instead of semicolon?
- L20: ungrammatical sentence
- Would be good to introduce the term "probe" somewhere in l.37-43 of the introduction (or earlier). You mention the "Probeless" classifier multiple times in the intro/abstract, but don't define this term anywhere.
- L102: represents --> represent
- L150: extra "i"
- L205-206 describe AvgOverlap as set-based (not taking ranking into account); this contradicts lines 222-223, where I think the word "ranking" is meant as "set". Can you clarify?
- I don't understand the role of Section 3.3 and its associated equation. Why wouldn't you just use Eqns. 7 and 8 with $|\mathcal{M}| = 2$?
- L255: the choice of hyperparameter values here seems like it could have a big impact on the results; did you do any search?
- Table 2 is never referred to in the text

**Limitations:**

Yes- see one suggestion above.

---

> ### Author Rebuttal · Authors · 2023-08-09
>
> **R: ….It would be nice for the authors to discuss limitations of the hypothesis…**
>
>
> A: We appreciate your comment. Given the variety of methods we considered, we anticipate discovering neurons with diverse properties, including polysemous characteristics. For instance, the ElasticNet regularization (Section 2.2.3) is theoretically capable of identifying both singular and multiple function neurons. We also found in the pairwise analysis (Figure 1) that neuron interpretation methods discover partly different neurons. In other words, the discoverability of a neuron varies depending on the neuron interpretation method used.
> We acknowledge the value of discussing these findings as it would offer insights into the potential limitations of each neuron interpretation method discussed in the paper and will provide future research directions. We have added the following limitations of interpretation methods in the paper.
> ```
> A limitation of current neuron interpretation methods is that they do not explicitly target the discovery of neurons of diverse nature such as polysemantic, and superposition. Theoretically, ElasticNet regularization is capable of discovering neurons learning a singular function and multiple functions. The other methods such as Probeless are incapable of discovering multifunction neurons. An explicit modeling of neurons of different nature in a neuron interpretation method may result in discovering novel sets of neurons.
> ```
>
> **R: The paper is lacking a related works section….**
>
>
> A: We agree that having a related work section will be useful to understand the work in the context of the other work on interpretability and neuron interpretation. We have added a related work section that provides a broad view of the field, clearly describes the scope of our paper and mentions other works on neuron interpretation. We have provided the related work section in the overall rebuttal.
>
>
> **R: I think it would be good … each method can be viewed as producing a ranking …**
>
>
> A: That’s a great suggestion. We will clarify this in the paper and explicitly mention that the evaluation metrics are designed in a way that won’t result in biased evaluation with respect to set-based and ranked-based methods.

---

> > ### Comment · Reviewer_z745 · 2023-08-14
> > **Response to Rebuttal**
> >
> > Thank you for the response. I appreciate the addition of the related works section. I will keep my score.

---

### Author Rebuttal · Authors · 2023-08-09

We thank the reviewers for their insightful comments, questions and suggestions. We have incorporated their suggestions, and answered specific concerns below each review. At a high level, we have significantly overhauled the related work (*included below for reviewers z745 and QcFR*), and defined the scope of our work within the field. We have run additional experiments to further confirm the results (*pdf attached*), as well as revamped the limitations section to incorporate the thoughtful suggestions by the reviewers. Following is the related work section that we have added to the paper.

```
Related Work
The area of interpreting deep learning models constitutes a broad expanse of research. This section provides a synthesized overview of diverse interpretability subareas within deep learning models applied to Natural Language Processing (NLP), while also outlining the scope of our study.
Attribution Methods Feature importance and attribution methods endeavor to identify the contribution of input features to predictions. These methodologies predominantly rely on the gradient of the output concerning the input feature and determine input feature importance by evaluating the magnitude of gradient values (Denil et al., 2014; Sundararajan et al., 2017). Please see Danilevsky et al. (2020) for a comprehensive survey.
Counterfactual Intervention revolves around an intricate analysis of the interplay between input features and predictions. This approach involves manipulating inputs and quantifying resulting output alterations. Diverse intervention strategies, including erasing input words, removing multiple input words, and substituting input words with different meanings, have been scrutinized (Li et al., 2016b; Ribeiro et al., 2018).
Attention Weights Numerous investigations have been directed towards interpreting components of deep learning models at varying levels of granularity. For instance, attention weights have emerged as a viable metric to gauge the interrelation between input instances and model outputs (Martins & Astudillo, 2016; Vig, 2019). Along these lines, Geva et al. (2021) delved into the analysis of feedforward neural network components within the transformer model, revealing their functionality as key-value memories. Additionally, Voita et al. (2019) demonstrated that pruning many attention heads has minimal impact on performance.
Mechanistic Interpretability puts a focus into the reverse engineering of network weights to comprehend their behavior. Building upon the Distill Circuits thread, Elhage et al. (2021) investigated two-layered transformer models with attention blocks, identifying attention heads contributing to in-context learning. This understanding was further extended to larger transformer-based language models by Olsson et al. (2022). To enhance neuron interpretability, Elhage et al. (2022) introduced a Softmax Linear unit as an activation function replacement. Wang et al. (2022) attempted to bridge mechanistic interpretability findings in small networks to large ones, particularly GPT-2 small. Their approach involved iteratively tracing influential model components from predictions using causal intervention. They showcased the potential of mechanistic interpretability in understanding extensive models, while also highlighting associated challenges.
Representation Analysis involves probing network representations concerning predefined concepts, particularly linguistic, to quantify the extent of knowledge captured in these representations Belinkov et al. (2017); Conneau et al. (2018); Liu et al. (2019a); Tenney et al. (2019). This is often realized through training diagnostic classifiers for specific concepts, wherein classifier accuracy serves as an indicator of concept knowledge within representations. See Belinkov & Glass (2019) for a comprehensive survey.
Neuron Interpretation A more intricate form of representation analysis, termed neuron interpretation, delves into how knowledge is organized within the network (Sajjad et al., 2022b). This approach establishes connections between neurons and predefined concepts, offering insights into where and how specific concept knowledge is assimilated. Work done on neuron analysis can be broadly classified into 3 groups: Neuron visualization involves manual identification of patterns across a set of sentences (Li et al., 2016a; Karpathy et al., 2015).More recently Foote et al. (2023) proposed an automated approach to enhance interpretability of Large Language Models (LLMs) by extracting and visualizing individual neuron behaviors as interpretable graphs. Corpus-based Methods explore the role of a neuron through techniques such as ranking sentences in a corpus Kádár et al. (2017b), generating synthetic sentences Poerner et al. (2018) that maximize its activation, or computing neuron-level statistics over a corpus Mu & Andreas (2020); Suau et al. (2020); Antverg & Belinkov (2022). (Bills et al., 2023) recently proposed an algorithm to generate neuron explanations, simulating activations using a simulator model (an LLM), and scoring the results. Probing Methods identify salient neurons for a concept by training a classifier using neuron activations as features Radford et al. (2019); Lakretz et al. (2019); Dalvi et al. (2019) or fitting a multivariate Gaussian over all neurons and then extracting individual probes for single neurons Torroba Hennigen et al. (2020). In this paper, we focus on the neuron interpretation methods that take a concept as input and find neurons with respect to the concept. We considered all methods mentioned in the recent survey on neuron interpretation Sajjad et al. (2022b) in our study. We propose an evaluation framework to formalize the evaluation and comparison of results across methods. Moreover, we propose a novel method, MeanSelect and present a case study of using the evaluation framework.
```

---

### Decision · Program_Chairs · 2023-09-21

**Decision:**

Accept (poster)

**Comment:**

The reviewers overall agree there is a significant contribution of neural interpretability. The authors have addressed reviewer concerns very well, but I highly recommend they ensure there are no further mistakes in their results or their presentation in the tables.